# Stochastic Douglas-Rachford Splitting for Regularized Empirical Risk Minimization: Convergence, Mini-batch, and Implementation

**Aysegul Bumin**                                             *aysegul.bumin@ufl.edu*
*Department of Computer and Information Science and Engineering*
*University of Florida*

**Kejun Huang**                                              *kejun.huang@ufl.edu*
*Department of Computer and Information Science and Engineering*
*University of Florida*

**Reviewed on OpenReview:** *https://openreview.net/forum?id=uvDD9rN6Zz*

## Abstract

In this paper, we study the stochastic Douglas-Rachford splitting (SDRS) for general empirical risk minimization (ERM) problems with regularization. Our first contribution is to prove its convergence for both convex and strongly convex problems; the convergence rates are $\mathcal{O}(1/\sqrt{t})$ and $\mathcal{O}(1/t)$, respectively. Since SDRS reduces to the stochastic proximal point algorithm (SPPA) when there is no regularization, it is pleasing to see the result matches that of SPPA, under the same mild conditions. We also propose the mini-batch version of SDRS that handles multiple samples simultaneously while maintaining the same efficiency as that of a single one, which is not a straight-forward extension in the context of stochastic proximal algorithms. We show that the mini-batch SDRS again enjoys the same convergence rate. Furthermore, we demonstrate that, for some of the canonical regularized ERM problems, each iteration of SDRS can be efficiently calculated either in closed form or in close to closed form via bisection—the resulting complexity is identical to, for example, the stochastic (sub)gradient method. Experiments on real data demonstrate its effectiveness in terms of convergence compared to SGD and its variants.

## 1 Introduction

Regularized empirical risk minimization (ERM) is the foundation for supervised machine learning (Vapnik, 1991). It formulates the learning problem as the following optimization problem

$$\underset{\boldsymbol{w} \in \mathbb{R}^d}{\text{minimize}} \quad \frac{1}{n} \sum_{i=1}^{n} f_i(\boldsymbol{w}) + g(\boldsymbol{w}), \tag{1}$$

where each $f_i$ is the loss function defined on a data sample over a training set, and $g$ is a regularization term to help improve the generalization ability of the model.

Large-scale machine learning models are ideally trained in a sample-by-sample manner in order to reduce computational and memory overhead. The majority of existing stochastic algorithms use the stochastic gradient descent (SGD) framework (Robbins & Monro, 1951; Bottou et al., 2018), possibly with an additional proximal step to handle the (often nonsmooth) regularization function $g$. Convergence is often $\mathcal{O}(1/\sqrt{t})$ for convex functions and $\mathcal{O}(1/t)$ for strongly convex functions. Some follow-up work focus on improving the theoretical convergence rate, such as SVRG (Johnson & Zhang, 2013), SDCA (Shalev-Shwartz & Zhang, 2013), SAGA (Defazio et al., 2014), and many more (Defazio, 2016; Lei et al., 2017; Schmidt et al., 2017;

Allen-Zhu, 2017; 2018a;b), at the expense of additional computational or memory overhead at the order of $n$. On the other hand, some practical variations have been proposed, such as AdaGrad (Duchi et al., 2011), Adam (Kingma & Ba, 2014), AdaBelief (Zhuang et al., 2020), and SGD-momentum (Liu et al., 2020), even though their theoretical convergence have not been shown to be better.

More recently, there emerges a family of stochastic algorithms that does not rely on just first-order derivatives but the full information of the stochastic functions (in some literature they are called model-based stochastic algorithms). This is most representative by the stochastic proximal point algorithm (SPPA) that tries to solve (1) without the regularization term $g(\boldsymbol{w})$ by the update $\boldsymbol{w}_{t+1} = \text{Prox}_{\lambda_t f_{i_t}}(\boldsymbol{w}_t)$, where $i_t$ is randomly sampled from the index set $\{1, \ldots, n\}$ (Bertsekas, 2011; Ryu & Boyd, 2014; Bianchi, 2016; Pătraşcu, 2020; Toulis et al., 2021; Bumin & Huang, 2021). Perhaps due to the fact that the algorithm relies on implementing the proximal operator of somewhat arbitrary functions, which could be as hard as solving the problem itself, follow-up works focus on variations of SPPA that typically involve a (sub)gradient update (Wang & Bertsekas, 2013; Duchi & Ruan, 2018; Asi & Duchi, 2019; Davis & Drusvyatskiy, 2019). As explained by Toulis et al. (2021) and later in this paper as well, in the context of stochastic optimization the pertinent proximal operators could be calculated efficiently, despite that they seem lack of structures to be exploited. There remains two challenges for applying SPPA in practice:

1. How to handle the regularization term in a more systematic manner? A naive approach would simply treat it as part of the stochastic function and cope with it stochastically, which is certainly not ideal. On the other hand, since most existing regularization terms have efficient proximal operators, it does not seem appropriate to use gradient-based update to handle it as suggested by some of the SPPA variants such as (Wang & Bertsekas, 2013; Duchi & Ruan, 2018; Asi & Duchi, 2019; Davis & Drusvyatskiy, 2019).
2. How to allow the algorithm to do mini-batch, i.e., using more than a single data sample to apply the updates? This is trivial for SGD-based methods since any (sub-)gradient is additive, so their stochastic gradients can be calculated in parallel and summed together. For SPPA, however, mini-batch seems not only complicated but also hard to parallelize. One attempt was made in (Chadha et al., 2022) but is parallelizable only if the approximation of the proximal update boils down to a stochastic gradient step.

In this paper, we address the two issues with similar convergence guarantees. First we revisit the stochastic Douglas-Rachford splitting (SDRS) to solve (1) with the regularization term. Then we propose SDRS mini-batch, which is inspired by consensus Douglas-Rachford. The mini-batch version of SDRS also provides a principled way of doing SPPA mini-batch that can be fully parallelized by essentially dropping the regularization term. We show that for both cases the convergence guarantee of SDRS is exactly the same as that of SPPA, i.e., with rate $\mathcal{O}(1/\sqrt{t})$ for convex problems and $\mathcal{O}(1/t)$ for strongly convex problems.

## 1.1 Stochastic Douglas-Rachford splitting (SDRS)

The stochastic Douglas-Rachford splitting (SDRS) is described in Algorithm 1. As we can see, the algorithm involves two proximal operators in each iteration. The proximal operator of a function $f$ at point $\tilde{\boldsymbol{w}}$ is defined as $\text{Prox}_f(\tilde{\boldsymbol{w}}) = \arg\min_{\boldsymbol{w}} f(\boldsymbol{w}) + (1/2)\|\boldsymbol{w} - \tilde{\boldsymbol{w}}\|^2$. Comparing SDRS with some well-known stochastic algorithm, we notice that:

- if line 4 is replaced by a simple stochastic gradient update, then it becomes the proximal SGD algorithm;
- if there is no regularization $g$, line 2 becomes $\boldsymbol{w}_{t+1} = \tilde{\boldsymbol{w}}_t$, and the algorithm becomes the stochastic proximal point algorithm (SPPA) (Rockafellar, 1976);
- if the stochastic step in line 3 is deterministic, then it is the (deterministic) Douglas-Rachford splitting (Lions & Mercier, 1979; Eckstein & Bertsekas, 1992), which is essentially equivalent to the celebrated alternating direction method of multipliers (Gabay & Mercier, 1976; Boyd et al., 2011).

There have been attempts to analyze the convergence of SDRS (Shi & Liu, 2016; Salim et al., 2018), but they both make the assumption that $f_i$ are differentiable, which is not ideal since differentiability is typically not required by any proximal algorithm. We will provide convergence analysis for SDRS that does not require any

---

**Algorithm 1** Stochastic Douglas-Rachford splitting Algorithm (SDRS)

---
1: initialize $\tilde{\boldsymbol{w}}_0$
2: **for** $t = 0, 1, \ldots, T$ **do**
3:    $\boldsymbol{w}_{t+1} \leftarrow \text{Prox}_{\lambda_t g}(\tilde{\boldsymbol{w}}_t)$
4:    randomly draw $i_t$ uniformly from $\{1, \ldots, n\}$
5:    $\tilde{\boldsymbol{w}}_{t+1} \leftarrow \tilde{\boldsymbol{w}}_t + \text{Prox}_{\lambda_t f_{i_t}}(2\boldsymbol{w}_{t+1} - \tilde{\boldsymbol{w}}_t) - \boldsymbol{w}_{t+1}$
6: **end for**
7: **return** $\hat{\boldsymbol{w}} = (\sum_{t=1}^T \lambda_t \boldsymbol{w}_t) / \sum_{t=1}^T \lambda_t$

---

function to be differentiable, as well as several additional results such as convergence under strong convexity and a mini-batch extension.

## 1.2   SDRS with mini-batch

One of the important reasons why SGD-type algorithms are so widely used is that it effortlessly include multiple samples in one stochastic update, called mini-batch. Due to linearity of differentiation, the mini-batch stochastic gradient is simply the average of the $f_i$'s that are sampled. Furthermore, each stochastic gradient can be computed in parallel, and a central node only need to collect their average, making it extremely easy to parallelize.

For SDRS, if each stochastic update involve multiple data samples, evaluating the proximal operator becomes significantly more computationally demanding. Suppose we consider $p$ samples in each stochastic step, not only is the complexity be in general $\mathcal{O}(p^2 d + p^3)$, but it is also hard to exploit parallelization. To address this issue, we propose Algorithm 2 that is inspired by consensus optimization using Douglas-Rachford splitting, which generates $p+1$ sequences $\{\tilde{\boldsymbol{w}}_t^{(1)}\}, \ldots, \{\tilde{\boldsymbol{w}}_t^{(p)}\}$, and $\{\boldsymbol{w}_t\}$. At iteration $t$, $i_t^{(k)}$ is the sample index obtained by processor $k$; in other words, each processor is in charge of computing only $\text{Prox}_{\lambda_t f_{i_t^{(k)}}}$, which is assumed to be easy to compute as will be discussed in §3. Then the central node collects their average and computes $\text{Prox}_{\lambda_t g}$. The deterministic counterpart of Algorithm 2 is when $p = n$ and each $i_t^{(k)} = k$, which would be equivalent to consensus ADMM (Boyd et al., 2011) after reorganization of the algorithm.

---

**Algorithm 2** Stochastic Douglas-Rachford splitting Algorithm (SDRS) with mini-batch

---
1: initialize $\tilde{\boldsymbol{w}}_0^{(1)} = \cdots = \tilde{\boldsymbol{w}}_0^{(p)} = \tilde{\boldsymbol{w}}_0$
2: **for** $t = 0, 1, \ldots, T$ **do**
3:    $\boldsymbol{w}_{t+1} \leftarrow \text{Prox}_{\lambda_t g}\left(\dfrac{1}{p} \sum_{k=1}^p \tilde{\boldsymbol{w}}_t^{(k)}\right)$
4:    **for** $k = 1, \ldots, p$ **do**
5:       randomly draw $i_t^{(k)}$ uniformly from $\{1, \ldots, n\}$
6:       $\tilde{\boldsymbol{w}}_{t+1}^{(k)} \leftarrow \tilde{\boldsymbol{w}}_t^{(k)} + \text{Prox}_{\lambda_t f_{i_t^{(k)}}}\left(2\boldsymbol{w}_{t+1} - \tilde{\boldsymbol{w}}_t^{(k)}\right) - \boldsymbol{w}_{t+1}$
7:    **end for**
8: **end for**
9: **return** $\hat{\boldsymbol{w}} = (\sum_{t=1}^T \lambda_t \boldsymbol{w}_t) / \sum_{t=1}^T \lambda_t$

---

An interesting observation is that if Problem (1) does not involve a regularization term $g$, then line 2 of the algorithm reduces to a simple averaging of the individual proximal outputs. This also suggests that one way of doing mini-batch for SPPA should be Algorithm 2 with line 2 being the average, which is not the same as any of the proposed mini-batch methods in (Chadha et al., 2022). The mini-batch approach proposed by (Chadha et al., 2022) allows approximate proximal operators, but is fully parallelizable only if the approximation reduces to a stochastic gradient update. Our proposed mini-batch is the first variant of SDRS and SPPA that is fully proximal and parallelizable.

## 2 Convergence Analysis

In this section, we provide convergence analysis of SDRS for general convex loss functions with a regularizer (1) to a global minimum in expectation. In recent years there have been some work tackling the convergence analysis of stochastic proximal point algorithms, e.g., (Bertsekas, 2011; Pătraşcu, 2020; Toulis et al., 2021) and stochastic Douglas-Rachford splitting (Shi & Liu, 2016; Salim et al., 2018). The latter two are the most relevant work, but they both make additional assumptions that $f_i$ are differentiable, which is uncommon for proximal algorithms. Our result resembles that of SPPA by Bertsekas (2011), which is satisfying since Douglas-Rachford splitting can be considered a natural extension to the proximal point algorithm to combine two or more proximal operators.

In what follows, we make the assumption that each $f_i(\cdot)$ is Lipschitz continuous with constant at least $L$, i.e.:

**Assumption 2.1.** There is a constant $L$ such that for all, $\boldsymbol{w}, \tilde{\boldsymbol{w}}$, and $f_i(\cdot)$, $|f_i(\boldsymbol{w}) - f_i(\tilde{\boldsymbol{w}})| \le L\|\boldsymbol{w} - \tilde{\boldsymbol{w}}\|$.

We note that this assumption is equivalent to the common assumption in analyses of stochastic (sub)gradient methods that all stochastic (sub)gradients are upperbounded (Vandenberghe, 2020, pp. 3.3). This is also the only assumption (other than convexity) used in establishing the convergence of SPPA (and some of its variants) (Bertsekas, 2011). Since the $g$ function is not handled in a stochastic fashion, it is pleasing to see that it is not required to satisfy the Lipschitz continuous assumption.

For simplicity, in the main paper we provide convergence analysis *in expectation*. In the appendix we utilize the supermartingale convergence theorem to establish convergence *in probability*. However, the key inequalities are established in this section, and the rest of the steps are somewhat standardized, which we relegate to the appendix.

### 2.1 Generic convex case without strong convexity

In this subsection we show that for general convex functions that are not necessarily strongly convex, SDRS converges in expectation with rate $\mathcal{O}(1/\sqrt{t})$. This is the same rate as SPPA under the same assumption (Bertsekas, 2011), as well as SGD under the same assumption (Bottou et al., 2018).

**Theorem 2.2.** *Suppose all $f_1, \ldots, f_n$ and $g$ are convex, and Assumption 2.1 holds. If a solution $\boldsymbol{w}_\star$ exists, then with initialization $\tilde{\boldsymbol{w}}_0$, the the sequence $\boldsymbol{w}_1, \ldots, \boldsymbol{w}_T$ generated by SDRS (Algorithm 1) satisfies*

$$\mathrm{E}\left[\frac{1}{n}\sum_{i=1}^n f_i(\hat{\boldsymbol{w}}) + g(\hat{\boldsymbol{w}})\right] - \left(\frac{1}{n}\sum_{i=1}^n f_i(\boldsymbol{w}_\star) + g(\boldsymbol{w}_\star)\right) \le \frac{\|\tilde{\boldsymbol{w}}_0 - \boldsymbol{w}_\star\|^2 + \sum_{t=1}^T \lambda_t^2 L^2}{2\sum_{t=1}^T \lambda_t} \tag{2}$$

*Proof.* As per the update rules defined in Algorithm 1, we have that

$$\frac{1}{\lambda_t}(\tilde{\boldsymbol{w}}_t - \boldsymbol{w}_{t+1}) \in \partial g(\boldsymbol{w}_{t+1}) \quad \text{and} \quad \frac{1}{\lambda_t}(\boldsymbol{w}_{t+1} - \tilde{\boldsymbol{w}}_{t+1}) \in \partial f_{i_t}(\tilde{\boldsymbol{w}}_{t+1} - \tilde{\boldsymbol{w}}_t + \boldsymbol{w}_{t+1}).$$

Since $g$ and $f_{i_t}$ are convex, their first-order conditions imply

$$g(\boldsymbol{w}_\star) \ge g(\boldsymbol{w}_{t+1}) + \frac{1}{\lambda_t}(\tilde{\boldsymbol{w}}_t - \boldsymbol{w}_{t+1})^\top(\boldsymbol{w}_\star - \boldsymbol{w}_{t+1}) \tag{3}$$

$$f_{i_t}(\boldsymbol{w}_\star) \ge f_{i_t}(\tilde{\boldsymbol{w}}_{t+1} - \tilde{\boldsymbol{w}}_t + \boldsymbol{w}_{t+1}) + \frac{1}{\lambda_t}(\boldsymbol{w}_{t+1} - \tilde{\boldsymbol{w}}_{t+1})^\top(\boldsymbol{w}_\star - \tilde{\boldsymbol{w}}_{t+1} + \tilde{\boldsymbol{w}}_t - \boldsymbol{w}_{t+1}). \tag{4}$$

Adding (3) and (4) together and rearrange, we have

$$f_{i_t}(\tilde{\boldsymbol{w}}_{t+1} - \tilde{\boldsymbol{w}}_t + \boldsymbol{w}_{t+1}) + g(\boldsymbol{w}_{t+1}) - f_{i_t}(\boldsymbol{w}_\star) - g(\boldsymbol{w}_\star)$$

$$\leq \frac{1}{\lambda_t}(\boldsymbol{w}_{t+1} - \tilde{\boldsymbol{w}}_t)^\top(\boldsymbol{w}_\star - \boldsymbol{w}_{t+1}) + \frac{1}{\lambda_t}(\tilde{\boldsymbol{w}}_{t+1} - \boldsymbol{w}_{t+1})^\top(\boldsymbol{w}_\star - \tilde{\boldsymbol{w}}_{t+1} + \tilde{\boldsymbol{w}}_t - \boldsymbol{w}_{t+1})$$

$$= \frac{1}{\lambda_t}(\tilde{\boldsymbol{w}}_{t+1} - \tilde{\boldsymbol{w}}_t)^\top(\boldsymbol{w}_\star - \boldsymbol{w}_{t+1}) + \frac{1}{\lambda_t}(\tilde{\boldsymbol{w}}_{t+1} - \boldsymbol{w}_{t+1})^\top(\tilde{\boldsymbol{w}}_t - \tilde{\boldsymbol{w}}_{t+1})$$

$$= \frac{1}{\lambda_t}(\tilde{\boldsymbol{w}}_{t+1} - \tilde{\boldsymbol{w}}_t)^\top(\boldsymbol{w}_\star - \tilde{\boldsymbol{w}}_{t+1})$$

$$= \frac{1}{2\lambda_t}\|\tilde{\boldsymbol{w}}_t - \boldsymbol{w}_\star\|^2 - \frac{1}{2\lambda_t}\|\tilde{\boldsymbol{w}}_{t+1} - \boldsymbol{w}_\star\|^2 - \frac{1}{2\lambda_t}\|\tilde{\boldsymbol{w}}_t - \tilde{\boldsymbol{w}}_{t+1}\|^2. \tag{5}$$

We now invoke Assumption 2.1 to have

$$f_{i_t}(\boldsymbol{w}_{t+1}) - L\|\tilde{\boldsymbol{w}}_t - \tilde{\boldsymbol{w}}_{t+1}\| \leq f_{i_t}(\tilde{\boldsymbol{w}}_{t+1} - \tilde{\boldsymbol{w}}_t + \boldsymbol{w}_{t+1}). \tag{6}$$

Combining (5) and (6), we have

$$2\lambda_t\left(f_{i_t}(\boldsymbol{w}_{t+1}) + g(\boldsymbol{w}_{t+1}) - f_{i_t}(\boldsymbol{w}_\star) - g(\boldsymbol{w}_\star)\right) \leq$$
$$\|\tilde{\boldsymbol{w}}_t - \boldsymbol{w}_\star\|^2 - \|\tilde{\boldsymbol{w}}_{t+1} - \boldsymbol{w}_\star\|^2 - \|\tilde{\boldsymbol{w}}_t - \tilde{\boldsymbol{w}}_{t+1}\|^2 + 2\lambda_t L\|\tilde{\boldsymbol{w}}_t - \tilde{\boldsymbol{w}}_{t+1}\|.$$

Furthermore, we notice that

$$-\|\tilde{\boldsymbol{w}}_t - \tilde{\boldsymbol{w}}_{t+1}\|^2 + 2\lambda_t L\|\tilde{\boldsymbol{w}}_t - \tilde{\boldsymbol{w}}_{t+1}\| = -\left(\|\tilde{\boldsymbol{w}}_t - \tilde{\boldsymbol{w}}_{t+1}\|^2 - \lambda_t L\right)^2 + \lambda_t^2 L^2 \leq \lambda_t^2 L^2.$$

As a result,

$$2\lambda_t\left(f_{i_t}(\boldsymbol{w}_{t+1}) + g(\boldsymbol{w}_{t+1}) - f_{i_t}(\boldsymbol{w}_\star) - g(\boldsymbol{w}_\star)\right) \leq \|\tilde{\boldsymbol{w}}_t - \boldsymbol{w}_\star\|^2 - \|\tilde{\boldsymbol{w}}_{t+1} - \boldsymbol{w}_\star\|^2 + \lambda_t^2 L^2. \tag{7}$$

Now we take conditional expectation of the random variable $i_t$, conditioned on $\boldsymbol{w}_{t+1}$ and $\tilde{\boldsymbol{w}}_t$; according to the update rule in Algorithm 1, only the value of $\tilde{\boldsymbol{w}}_{t+1}$—but not $\boldsymbol{w}_{t+1}$ or $\tilde{\boldsymbol{w}}_t$—depends on $i_t$, we obtain the inequality

$$2\lambda_t\left(\frac{1}{n}\sum_{i=1}^n f_i(\boldsymbol{w}_{t+1}) + g(\boldsymbol{w}_{t+1}) - \frac{1}{n}\sum_{i=1}^n f_i(\boldsymbol{w}_\star) - g(\boldsymbol{w}_\star)\right) \leq \|\tilde{\boldsymbol{w}}_t - \boldsymbol{w}_\star\|^2 - \mathrm{E}_{i_t}\|\tilde{\boldsymbol{w}}_{t+1} - \boldsymbol{w}_\star\|^2 + \lambda_t^2 L^2. \tag{8}$$

Taking total expectation of (8) and summing over the inequalities with $t = 0, 1, \ldots, T$, we have

$$\sum_{t=0}^T 2\lambda_t\left(\mathrm{E}\left[\frac{1}{n}\sum_{i=1}^n f_i(\boldsymbol{w}_{t+1}) + g(\boldsymbol{w}_{t+1})\right] - \left(\frac{1}{n}\sum_{i=1}^n f_i(\boldsymbol{w}_\star) + g(\boldsymbol{w}_\star)\right)\right)$$

$$\leq \|\tilde{\boldsymbol{w}}_0 - \boldsymbol{w}_\star\|^2 - \mathrm{E}\|\tilde{\boldsymbol{w}}_{T+1} - \boldsymbol{w}_\star\|^2 + \sum_{t=0}^T \lambda_t^2 L^2 \leq \|\tilde{\boldsymbol{w}}_0 - \boldsymbol{w}_\star\|^2 + \sum_{t=0}^T \lambda_t^2 L^2.$$

Dividing both sides by $2\sum_t \lambda_t$ and lowerbouding the left-hand-side by Jensen's inequality (since the objective function (1) is convex),

$$\frac{1}{n}\sum_{i=1}^n f_i(\hat{\boldsymbol{w}}) + g(\hat{\boldsymbol{w}}) \leq \frac{1}{\sum_{t=1}^T \lambda_t}\sum_{t=1}^T \lambda_t\left(\frac{1}{n}\sum_{i=1}^n f_i(\boldsymbol{w}_t) + g(\boldsymbol{w}_t)\right),$$

we obtain (2). □

**Remarks.** The proof is somewhat reminiscent to that of SPPA by Bertsekas (2011). Indeed, if function $g$ is absent, SDRS reduces to SPPA since $\boldsymbol{w}_{t+1} = \tilde{\boldsymbol{w}}_t$, and it is pleasing to see that the same result of SPPA holds for the more general SDRS. The fundamental observation in (Bertsekas, 2011) is that inequality (4) gives an upperbound on $f_{i_t}(\tilde{\boldsymbol{w}}_{t+1}) - f_{i_t}(\boldsymbol{w}_\star)$; however, when taking expectation over the random variable $i_t$, the variable $\tilde{\boldsymbol{w}}_{t+1}$ also depends on $i_t$, which would not lead to a meaningful expression. This is why we need to invoke Assumption 2.1 to bound $f_{i_t}(\tilde{\boldsymbol{w}}_t) - f_{i_t}(\boldsymbol{w}_\star)$ instead. In the case of SDRS in (5), not only is $f_{i_t}$ evaluated at a point that is dependent on $i_t$, but also $f_{i_t}$ and $g$ are evaluated at different points; when changing the argument of $f_{i_t}$ to be the same as $g$, we also managed to make the argument independent of $i_t$, thus similar steps can be made in the sequel.

Theorem 2.2 establishes inequality (2) for any choice of $\lambda_t$. We now detail some specific choices of the step sizes. The basic arguments are essentially the same as those that have been well-studied for SGD.

**Corollary 2.3** (Diminishing step sizes). *Suppose all $f_1, \ldots, f_n$ and $g$ are convex, and Assumption 2.1 holds. If a solution $\boldsymbol{w}_\star$ exists, then with initialization $\tilde{\boldsymbol{w}}_0$, the sequence $\boldsymbol{w}_1, \ldots, \boldsymbol{w}_T$ generated by SDRS (Algorithm 1) with diminishing step sizes such that $\lambda_t \to 0$ and $\sum_{t=1}^{\infty} \lambda_t = \infty$ satisfies*

$$\mathrm{E}\left[\frac{1}{n}\sum_{i=1}^{n} f_i(\hat{\boldsymbol{w}}) + g(\hat{\boldsymbol{w}})\right] \to \frac{1}{n}\sum_{i=1}^{n} f_i(\boldsymbol{w}_\star) + g(\boldsymbol{w}_\star). \tag{9}$$

*Proof.* Suppose the step sizes are square summable, i.e., $\sum_t \gamma^{(t)2} < \infty$, then it is obvious that on the right-hand-side of (2), the numerator is finite while the denominator goes to infinity, which proves the result. Even if the step sizes are not square summable either, it can be shown that the quotient $\sum_t \lambda_t^2 / \sum_t \lambda_t \to 0$. A proof can be found in the supplementary, even though it is not new. □

Corollary 2.3 suggests that some well-known diminishing rules such as $\lambda_t = \tau/t$ or $\lambda_t = \tau/\sqrt{t}$ all guarantee expected convergence of SDRS.

**Corollary 2.4** (Constant step sizes). *Suppose all $f_1, \ldots, f_n$ and $g$ are convex, and Assumption 2.1 holds. If a solution $\boldsymbol{w}_\star$ exists, then with initialization $\tilde{\boldsymbol{w}}_0$, the sequence $\boldsymbol{w}_1, \ldots, \boldsymbol{w}_T$ generated by SDRS (Algorithm 1) with constant step size $\lambda_t = \lambda$ satisfies*

$$\mathrm{E}\left[\frac{1}{n}\sum_{i=1}^{n} f_i(\hat{\boldsymbol{w}}) + g(\hat{\boldsymbol{w}})\right] - \left(\frac{1}{n}\sum_{i=1}^{n} f_i(\boldsymbol{w}_\star) + g(\boldsymbol{w}_\star)\right) \leq \frac{\|\tilde{\boldsymbol{w}}_0 - \boldsymbol{w}_\star\|^2}{2T\lambda} + \frac{L^2}{2}\lambda. \tag{10}$$

*This means:*

- *Letting $T \to \infty$, the expected optimality gap is upperbounded by a constant times $\lambda$.*

- *For a given $T$, the right-hand-side is minimized by letting $\lambda = \|\tilde{\boldsymbol{w}}_0 - \boldsymbol{w}_\star\|/L\sqrt{T}$, and substituting it back in (10) shows*

$$\mathrm{E}\left[\frac{1}{n}\sum_{i=1}^{n} f_i(\hat{\boldsymbol{w}}) + g(\hat{\boldsymbol{w}})\right] - \left(\frac{1}{n}\sum_{i=1}^{n} f_i(\boldsymbol{w}_\star) + g(\boldsymbol{w}_\star)\right) \leq \frac{\|\tilde{\boldsymbol{w}}_0 - \boldsymbol{w}_\star\|L}{\sqrt{T}}.$$

The proof is straight-forward and thus omitted. This shows that the expected convergence rate is $\mathcal{O}(1/\sqrt{t})$, which is the same as SPPA (Bertsekas, 2011) and SGD.

## 2.2 Strongly convex case

We now provide improved convergence analysis when $g$ is strongly convex. This is particularly useful when the regularization function is the Tikhonov regularization (Euclidean norm squared).

**Proposition 2.5.** *If all $f_1, \ldots, f_n$ are convex, $g$ is strongly convex with parameter $\mu$, and Assumption 2.1 holds, then with initialization $\tilde{\boldsymbol{w}}_0$, the sequence $\boldsymbol{w}_1, \ldots, \boldsymbol{w}_T$ generated by SDRS (Algorithm 1) with a constant step size $\lambda$ satisfies*

$$\mathrm{E}\left[\frac{1}{n}\sum_{i=1}^{n} f_i(\hat{\boldsymbol{w}}) + g(\hat{\boldsymbol{w}})\right] - \left(\frac{1}{n}\sum_{i=1}^{n} f_i(\boldsymbol{w}_\star) + g(\boldsymbol{w}_\star)\right) \leq \frac{\mu\|\tilde{\boldsymbol{w}}_0 - \boldsymbol{w}_\star\|^2}{(1+\lambda\mu/2)^T - 1} + \frac{\lambda L^2 + \lambda^2\mu L^2}{2}. \quad (11)$$

*This means SDRS converges, in expectation, linearly to a suboptimal point with gap proportional to $\lambda + \mu\lambda^2$. If $\lambda$ is small, which is usually the case in practice, then the gap is dominated by a constant times $\lambda$.*

*Proof.* Similar to the first few steps of the proof of Theorem 2.2, but replacing (3) with the strongly convex form

$$g(\boldsymbol{w}_\star) \geq g(\boldsymbol{w}_{t+1}) + \frac{1}{\lambda_t}(\tilde{\boldsymbol{w}}_t - \boldsymbol{w}_{t+1})^\top(\boldsymbol{w}_\star - \boldsymbol{w}_{t+1}) + \frac{\mu}{2}\|\boldsymbol{w}_\star - \boldsymbol{w}_{t+1}\|^2,$$

we get

$$2\lambda_t\left(f_{i_t}(\boldsymbol{w}_{t+1}) + g(\boldsymbol{w}_{t+1}) - f_{i_t}(\boldsymbol{w}_\star) - g(\boldsymbol{w}_\star)\right) \leq \|\tilde{\boldsymbol{w}}_t - \boldsymbol{w}_\star\|^2 - \|\tilde{\boldsymbol{w}}_{t+1} - \boldsymbol{w}_\star\|^2 + \lambda_t^2 L^2 - \lambda_t\mu\|\boldsymbol{w}_\star - \boldsymbol{w}_{t+1}\|^2.$$

Using the inequality $\|\boldsymbol{a} + \boldsymbol{b}\|^2 \leq 2\|\boldsymbol{a}\|^2 + 2\|\boldsymbol{b}\|^2$, we can further upperbound the right-hand-side by

$$2\lambda_t\left(f_{i_t}(\boldsymbol{w}_{t+1}) + g(\boldsymbol{w}_{t+1}) - f_{i_t}(\boldsymbol{w}_\star) - g(\boldsymbol{w}_\star)\right)$$

$$\leq \|\tilde{\boldsymbol{w}}_t - \boldsymbol{w}_\star\|^2 - \|\tilde{\boldsymbol{w}}_{t+1} - \boldsymbol{w}_\star\|^2 + \lambda_t^2 L^2 - \frac{\lambda_t\mu}{2}\|\boldsymbol{w}_\star - \tilde{\boldsymbol{w}}_{t+1}\|^2 + \lambda_t\mu\|\tilde{\boldsymbol{w}}_{t+1} - \boldsymbol{w}_{t+1}\|^2$$

$$\leq \|\tilde{\boldsymbol{w}}_t - \boldsymbol{w}_\star\|^2 - \left(1 + \frac{\lambda_t\mu}{2}\right)\|\tilde{\boldsymbol{w}}_{t+1} - \boldsymbol{w}_\star\|^2 + \lambda_t^2 L^2 + \lambda_t^3\mu L^2,$$

where we upperbound $\|\tilde{\boldsymbol{w}}_{t+1} - \boldsymbol{w}_{t+1}\|^2 \leq \lambda_t^2 L^2$ because $(1/\lambda_t)(\tilde{\boldsymbol{w}}_{t+1} - \boldsymbol{w}_{t+1})$ is a subgradient at $f_{i_t}(\tilde{\boldsymbol{w}}_{t+1} - \tilde{\boldsymbol{w}}_t + \boldsymbol{w}_{t+1})$, of which the norm is no more than $L$ due to Assumption 2.1.

Now we take conditional expectation of the random variable $i_t$, conditioned on $\boldsymbol{w}_{t+1}$ and $\tilde{\boldsymbol{w}}_t$; according to the update rule in Algorithm 1, only the value of $\tilde{\boldsymbol{w}}_{t+1}$—but not $\boldsymbol{w}_{t+1}$ or $\tilde{\boldsymbol{w}}_t$—depends on $i_t$, we obtain

$$2\lambda_t\left(\frac{1}{n}\sum_{i=1}^{n} f_i(\boldsymbol{w}_{t+1}) + g(\boldsymbol{w}_{t+1}) - \frac{1}{n}\sum_{i=1}^{n} f_i(\boldsymbol{w}_\star) - g(\boldsymbol{w}_\star)\right)$$

$$\leq \|\tilde{\boldsymbol{w}}_t - \boldsymbol{w}_\star\|^2 - \left(1 + \frac{\lambda_t\mu}{2}\right)\mathrm{E}_{i_t}\|\tilde{\boldsymbol{w}}_{t+1} - \boldsymbol{w}_\star\|^2 + \lambda_t^2 L^2 + \lambda_t^3\mu L^2. \quad (12)$$

Letting the step sizes be constant $\lambda_t = \lambda$, we multiply both sides by $(1 + \lambda\mu/2)^t$, take total expectation, and sum over $t = 0, \ldots, T$ to get

$$2\lambda\sum_{t=0}^{T}\left(1 + \frac{\lambda\mu}{2}\right)^t\left(\mathrm{E}\left[\frac{1}{n}\sum_{i=1}^{n} f_i(\boldsymbol{w}_{t+1}) + g(\boldsymbol{w}_{t+1})\right] - \left(\frac{1}{n}\sum_{i=1}^{n} f_i(\boldsymbol{w}_\star) + g(\boldsymbol{w}_\star)\right)\right)$$

$$\leq \|\tilde{\boldsymbol{w}}_0 - \boldsymbol{w}_\star\|^2 + (\lambda^2 L^2 + \lambda^3\mu L^2)\sum_{t=1}^{T}\left(1 + \frac{\lambda\mu}{2}\right)^t.$$

Applying the geometric series $\sum_{t=0}^{T}(1 + \lambda\mu/2)^t = ((1 + \lambda\mu/2)^T - 1)/(\lambda\mu/2)$, and lowerbound all the expected function values on left-hand-side by Jensen's inequality, we get (11). □

We now show that with diminishing step sizes, the expected optimality gap converges to zero at $\mathcal{O}(1/t)$ rate.

**Theorem 2.6.** *If all $f_1, \ldots, f_n$ are convex, $g$ is strongly convex with parameter $\mu$, and Assumption 2.1 holds. If a solution $\boldsymbol{w}_\star$ exists, then with initialization $\tilde{\boldsymbol{w}}_0$, the sequence $\boldsymbol{w}_1, \ldots, \boldsymbol{w}_T$ generated by SDRS (Algorithm 1) with diminishing step sizes $\lambda_t = \beta/t$ where $\beta > 4/\mu$, then*

$$\mathrm{E}\left[\frac{1}{n}\sum_{i=1}^{n} f_i(\hat{\boldsymbol{w}}) + g(\hat{\boldsymbol{w}})\right] \leq \frac{2\|\tilde{\boldsymbol{w}}_0 - \boldsymbol{w}_\star\|^2}{\beta T(T+1)} + \frac{2L^2\beta(1 + \mu\beta)}{T+1}. \quad (13)$$

*When $t$ goes relatively large, the right-hand-side is dominated by the second term, which yields the $\mathcal{O}(1/t)$ convergence rate.*

*Proof.* Taking total expectation of (12) and plug in the step sizes $\lambda_t = \beta/t$, we have

$$\mathrm{E}\left[\frac{1}{n}\sum_{i=1}^{n}f_i(\boldsymbol{w}_{t+1}) + g(\boldsymbol{w}_{t+1}) - \frac{1}{n}\sum_{i=1}^{n}f_i(\boldsymbol{w}_\star) - g(\boldsymbol{w}_\star)\right]$$

$$\leq \frac{t}{2\beta}\,\mathrm{E}\,\|\tilde{\boldsymbol{w}}_t - \boldsymbol{w}_\star\|^2 - \frac{2t+\mu\beta}{4\beta}\,\mathrm{E}\,\|\tilde{\boldsymbol{w}}_{t+1} - \boldsymbol{w}_\star\|^2 + \frac{L^2\beta}{2t} + \frac{\beta^2\mu L^2}{2t^2}$$

$$\leq \frac{t}{2\beta}\,\mathrm{E}\,\|\tilde{\boldsymbol{w}}_t - \boldsymbol{w}_\star\|^2 - \frac{t+2}{2\beta}\,\mathrm{E}\,\|\tilde{\boldsymbol{w}}_{t+1} - \boldsymbol{w}_\star\|^2 + \frac{L^2\beta(1+\beta\mu)}{2t},$$

where the last inequality stems from $\mu\beta > 4$ and $1/t^2 < 1/t$ for $t \geq 1$. Multiplying both sides by $t+1$ yields

$$(t+1)\,\mathrm{E}\left[\frac{1}{n}\sum_{i=1}^{n}f_i(\boldsymbol{w}_{t+1}) + g(\boldsymbol{w}_{t+1}) - \frac{1}{n}\sum_{i=1}^{n}f_i(\boldsymbol{w}_\star) - g(\boldsymbol{w}_\star)\right]$$

$$\leq \frac{t(t+1)}{2\beta}\,\mathrm{E}\,\|\tilde{\boldsymbol{w}}_t - \boldsymbol{w}_\star\|^2 - \frac{(t+1)(t+2)}{2\beta}\,\mathrm{E}\,\|\tilde{\boldsymbol{w}}_{t+1} - \boldsymbol{w}_\star\|^2 + \frac{L^2\beta(1+\beta\mu)(t+1)}{2t}$$

$$\leq \frac{t(t+1)}{2\beta}\,\mathrm{E}\,\|\tilde{\boldsymbol{w}}_t - \boldsymbol{w}_\star\|^2 - \frac{(t+1)(t+2)}{2\beta}\,\mathrm{E}\,\|\tilde{\boldsymbol{w}}_{t+1} - \boldsymbol{w}_\star\|^2 + L^2\beta(1+\beta\mu),$$

where, again, the last inequality is due to $(t+1)/2t < 1$ for $t \geq 1$. Now let $t = 1, 2, \ldots, T$ and sum over all inequalities, we get

$$\sum_{t=1}^{T}t\,\mathrm{E}\left[\frac{1}{n}\sum_{i=1}^{n}f_i(\boldsymbol{w}_t) + g(\boldsymbol{w}_t) - \frac{1}{n}\sum_{i=1}^{n}f_i(\boldsymbol{w}_\star) - g(\boldsymbol{w}_\star)\right]$$

$$\leq \frac{1}{\beta}\,\mathrm{E}\,\|\tilde{\boldsymbol{w}}_1 - \boldsymbol{w}_\star\|^2 - \frac{T(T+1)}{2\beta}\,\mathrm{E}\,\|\tilde{\boldsymbol{w}}_T - \boldsymbol{w}_\star\|^2 + TL^2\beta(1+\beta\mu)$$

$$\leq \frac{1}{\beta}\,\mathrm{E}\,\|\tilde{\boldsymbol{w}}_1 - \boldsymbol{w}_\star\|^2 + TL^2\beta(1+\beta\mu).$$

Lowerbounding the left-hand-side by Jensen's inequality and dividing both sides by $1+2+\cdots+T = T(T+1)/2$ yields (13). $\square$

## 2.3 Convergence of mini-batch SDRS

**Theorem 2.7.** *The convergence of mini-batch SDRS as in Algorithm 2 is exactly the same as SDRS, i.e.,*

- *for generic convex functions, we have*

$$\mathrm{E}\left[\frac{1}{n}\sum_{i=1}^{n}f_i(\hat{\boldsymbol{w}}) + g(\hat{\boldsymbol{w}})\right] - \left(\frac{1}{n}\sum_{i=1}^{n}f_i(\boldsymbol{w}_\star) + g(\boldsymbol{w}_\star)\right) \leq \frac{\|\tilde{\boldsymbol{w}}_0 - \boldsymbol{w}_\star\|^2 + \sum_{t=1}^{T}\lambda_t^2 L^2}{2\sum_{t=1}^{T}\lambda_t}, \quad (14)$$

  *which leads to a convergence rate of $\mathcal{O}(1/\sqrt{t})$ with appropriately chosen step sizes;*

- *if $g$ is strongly convex with parameter $\mu$, then with diminishing step sizes $\lambda_t = \beta/t$ where $\beta > 4/\mu$, then*

$$\mathrm{E}\left[\frac{1}{n}\sum_{i=1}^{n}f_i(\hat{\boldsymbol{w}}) + g(\hat{\boldsymbol{w}})\right] \leq \frac{2\|\tilde{\boldsymbol{w}}_0 - \boldsymbol{w}_\star\|^2}{\beta T(T+1)} + \frac{2L^2\beta(1+\mu\beta)}{T+1}. \quad (15)$$

  *When $T$ goes relatively large, the right-hand-side is dominated by the second term, which yields the $\mathcal{O}(1/t)$ convergence rate.*

*Proof.* As per the update rules defined in Algorithm 2, we have that

$$\frac{1}{\lambda_t}\left(\frac{1}{p}\sum_{k=1}^{p}\tilde{\boldsymbol{w}}_t^{(k)} - \boldsymbol{w}_{t+1}\right) \in \partial g(\boldsymbol{w}_{t+1})$$

and

$$\frac{1}{\lambda_t}(\boldsymbol{w}_{t+1} - \tilde{\boldsymbol{w}}_{t+1}^{(k)}) \in \partial f_{i_t^{(k)}}(\tilde{\boldsymbol{w}}_{t+1}^{(k)} - \tilde{\boldsymbol{w}}_t^{(k)} + \boldsymbol{w}_{t+1}), k = 1,\ldots,p.$$

Due to (strong) convexity, their first-order conditions imply

$$g(\boldsymbol{w}_\star) \geq g(\boldsymbol{w}_{t+1}) + \frac{1}{\lambda_t}\left(\frac{1}{p}\sum_{k=1}^{p}\tilde{\boldsymbol{w}}_t^{(k)} - \boldsymbol{w}_{t+1}\right)^\top(\boldsymbol{w}_\star - \boldsymbol{w}_{t+1}) + \frac{\mu}{2}\|\boldsymbol{w}_\star - \boldsymbol{w}_{t+1}\|^2,$$

$$f_{i_t^{(k)}}(\boldsymbol{w}_\star) \geq f_{i_t^{(k)}}(\tilde{\boldsymbol{w}}_{t+1}^{(k)} - \tilde{\boldsymbol{w}}_t^{(k)} + \boldsymbol{w}_{t+1}) + \frac{1}{\lambda_t}(\boldsymbol{w}_{t+1} - \tilde{\boldsymbol{w}}_{t+1}^{(k)})^\top(\boldsymbol{w}_\star - \tilde{\boldsymbol{w}}_{t+1}^{(k)} + \tilde{\boldsymbol{w}}_t^{(k)} - \boldsymbol{w}_{t+1}), k = 1,\ldots,p.$$

Summing them up gives

$$2\lambda_t\left(\frac{1}{p}\sum_{k=1}^{p}f_{i_t^{(k)}}(\tilde{\boldsymbol{w}}_{t+1}^{(k)} - \tilde{\boldsymbol{w}}_t^{(k)} + \boldsymbol{w}_{t+1}) + g(\boldsymbol{w}_{t+1}) - \frac{1}{p}\sum_{k=1}^{p}f_{i_t^{(k)}}(\boldsymbol{w}_\star) - g(\boldsymbol{w}_\star)\right)$$

$$\leq 2\left(\boldsymbol{w}_{t+1} - \frac{1}{p}\sum_{k=1}^{p}\tilde{\boldsymbol{w}}_t^{(k)}\right)^\top(\boldsymbol{w}_\star - \boldsymbol{w}_{t+1}) + \frac{2}{p}\sum_{k=1}^{p}(\tilde{\boldsymbol{w}}_{t+1}^{(k)} - \boldsymbol{w}_{t+1})^\top(\boldsymbol{w}_\star - \tilde{\boldsymbol{w}}_{t+1}^{(k)} + \tilde{\boldsymbol{w}}_t^{(k)} - \boldsymbol{w}_{t+1})$$

$$- \mu\lambda_t\|\boldsymbol{w}_\star - \boldsymbol{w}_{t+1}\|^2$$

$$= \frac{2}{p}\sum_{k=1}^{p}\left(\tilde{\boldsymbol{w}}_{t+1}^{(k)} - \tilde{\boldsymbol{w}}_t^{(k)}\right)^\top(\boldsymbol{w}_\star - \boldsymbol{w}_{t+1}) - \frac{2}{p}\sum_{k=1}^{p}(\tilde{\boldsymbol{w}}_{t+1}^{(k)} - \boldsymbol{w}_{t+1})^\top(\tilde{\boldsymbol{w}}_{t+1}^{(k)} - \tilde{\boldsymbol{w}}_t^{(k)})$$

$$- \mu\lambda_t\|\boldsymbol{w}_\star - \boldsymbol{w}_{t+1}\|^2$$

$$= \frac{2}{p}\sum_{k=1}^{p}\left(\tilde{\boldsymbol{w}}_{t+1}^{(k)} - \tilde{\boldsymbol{w}}_t^{(k)}\right)^\top(\boldsymbol{w}_\star - \tilde{\boldsymbol{w}}_{t+1}^{(k)}) - \mu\lambda_t\|\boldsymbol{w}_\star - \boldsymbol{w}_{t+1}\|^2$$

$$= \frac{1}{p}\sum_{k=1}^{p}\left(\|\boldsymbol{w}_\star - \tilde{\boldsymbol{w}}_t^{(k)}\|^2 - \|\boldsymbol{w}_\star - \tilde{\boldsymbol{w}}_{t+1}^{(k)}\|^2 - \|\tilde{\boldsymbol{w}}_{t+1}^{(k)} - \tilde{\boldsymbol{w}}_t^{(k)}\|^2\right) - \mu\lambda_t\|\boldsymbol{w}_\star - \boldsymbol{w}_{t+1}\|^2 \qquad (16)$$

The rest of the steps follows almost identical to those in the proof of Theorem 2.2 after (5). Invoking Assumption 2.1 to each $f_{i_t^{(k)}}$, we have

$$f_{i_t^{(k)}}(\boldsymbol{w}_{t+1}) - L\|\tilde{\boldsymbol{w}}_{t+1}^{(k)} - \tilde{\boldsymbol{w}}_t^{(k)}\| \leq f_{i_t^{(k)}}(\tilde{\boldsymbol{w}}_{t+1}^{(k)} - \tilde{\boldsymbol{w}}_t^{(k)} + \boldsymbol{w}_{t+1}), \quad k = 1,\ldots,p.$$

Plugging it into (16) gives

$$2\lambda_t\left(\frac{1}{p}\sum_{k=1}^{p}f_{i_t^{(k)}}(\boldsymbol{w}_{t+1}) + g(\boldsymbol{w}_{t+1}) - \frac{1}{p}\sum_{k=1}^{p}f_{i_t^{(k)}}(\boldsymbol{w}_\star) - g(\boldsymbol{w}_\star)\right)$$

$$\leq \frac{1}{p}\sum_{k=1}^{p}\left(\|\boldsymbol{w}_\star - \tilde{\boldsymbol{w}}_t^{(k)}\|^2 - \|\boldsymbol{w}_\star - \tilde{\boldsymbol{w}}_{t+1}^{(k)}\|^2 - \|\tilde{\boldsymbol{w}}_{t+1}^{(k)} - \tilde{\boldsymbol{w}}_t^{(k)}\|^2 + 2\lambda_t\|\tilde{\boldsymbol{w}}_{t+1}^{(k)} - \tilde{\boldsymbol{w}}_t^{(k)}\|\right) - \mu\lambda_t\|\boldsymbol{w}_\star - \boldsymbol{w}_{t+1}\|^2.$$

Also notice that

$$-\|\tilde{\boldsymbol{w}}_{t+1}^{(k)} - \tilde{\boldsymbol{w}}_t^{(k)}\|^2 + 2\lambda_t\|\tilde{\boldsymbol{w}}_{t+1}^{(k)} - \tilde{\boldsymbol{w}}_t^{(k)}\| = -\left(\|\tilde{\boldsymbol{w}}_{t+1}^{(k)} - \tilde{\boldsymbol{w}}_t^{(k)}\| - \lambda_t L\right)^2 + \lambda_t^2 L^2 \leq \lambda_t^2 L^2,$$

This implies

$$2\lambda_t\left(\frac{1}{p}\sum_{k=1}^{p}f_{i_t^{(k)}}(\boldsymbol{w}_{t+1}) + g(\boldsymbol{w}_{t+1}) - \frac{1}{p}\sum_{k=1}^{p}f_{i_t^{(k)}}(\boldsymbol{w}_\star) - g(\boldsymbol{w}_\star)\right)$$

$$\leq \frac{1}{p}\sum_{k=1}^{p}\left(\|\boldsymbol{w}_\star - \tilde{\boldsymbol{w}}_t^{(k)}\|^2 - \|\boldsymbol{w}_\star - \tilde{\boldsymbol{w}}_{t+1}^{(k)}\|^2\right) + \lambda_t^2 L^2 - \mu\lambda_t\|\boldsymbol{w}_\star - \boldsymbol{w}_{t+1}\|^2.$$

Using the inequality $\|\boldsymbol{a} + \boldsymbol{b}\|^2 \leq 2\|\boldsymbol{a}\|^2 + 2\|\boldsymbol{b}\|^2$, we can further upperbound the right-hand-side by

$$2\lambda_t \left( \frac{1}{p} \sum_{k=1}^p f_{i_t^{(k)}}(\boldsymbol{w}_{t+1}) + g(\boldsymbol{w}_{t+1}) - \frac{1}{p} \sum_{k=1}^p f_{i_t^{(k)}}(\boldsymbol{w}_\star) - g(\boldsymbol{w}_\star) \right)$$

$$\leq \frac{1}{p} \sum_{k=1}^p \left( \|\boldsymbol{w}_\star - \tilde{\boldsymbol{w}}_t^{(k)}\|^2 - \|\boldsymbol{w}_\star - \tilde{\boldsymbol{w}}_{t+1}^{(k)}\|^2 \right) + \lambda_t^2 L^2 - \frac{\mu \lambda_t}{p} \sum_{k=1}^p \left( \frac{1}{2} \|\boldsymbol{w}_\star - \tilde{\boldsymbol{w}}_{t+1}^{(k)}\|^2 - \|\tilde{\boldsymbol{w}}_{t+1}^{(k)} - \boldsymbol{w}_{t+1}\|^2 \right)$$

$$\leq \frac{1}{p} \sum_{k=1}^p \left( \|\boldsymbol{w}_\star - \tilde{\boldsymbol{w}}_t^{(k)}\|^2 - \left(1 + \frac{\lambda_t \mu}{2}\right) \|\boldsymbol{w}_\star - \tilde{\boldsymbol{w}}_{t+1}^{(k)}\|^2 \right) + \lambda_t^2 L^2 + \lambda_t^3 \mu L^2. \tag{17}$$

The next step is to take conditional expectation of the random variables $i_t^{(k)}$ in (17), conditioned on $\boldsymbol{w}_{t+1}$ and each of $\tilde{\boldsymbol{w}}_t^{(k)}$; according to the update rule in Algorithm 2, only the values of $\tilde{\boldsymbol{w}}_{t+1}^{(k)}$—but not $\boldsymbol{w}_{t+1}$ or any of $\tilde{\boldsymbol{w}}_t^{(k)}$—depends on $i_t^{(k)}$. Specifically, on the left-hand-side, we have that for each $i_t^{(k)}$:

$$\mathrm{E}_{i_t^{(k)}} f_{i_t^{(k)}}(\boldsymbol{w}_{t+1}) = \frac{1}{n} \sum_{i=1}^n f_i(\boldsymbol{w}_{t+1}) \quad \text{and} \quad \mathrm{E}_{i_t^{(k)}} f_{i_t^{(k)}}(\boldsymbol{w}_\star) = \frac{1}{n} \sum_{i=1}^n f_i(\boldsymbol{w}_\star),$$

so

$$2\lambda_t \left( \frac{1}{n} \sum_{i=1}^n f_i(\boldsymbol{w}_{t+1}) + g(\boldsymbol{w}_{t+1}) - \frac{1}{n} \sum_{i=1}^n f_i(\boldsymbol{w}_\star) - g(\boldsymbol{w}_\star) \right)$$

$$\leq \frac{1}{p} \sum_{k=1}^p \left( \|\boldsymbol{w}_\star - \tilde{\boldsymbol{w}}_t^{(k)}\|^2 - \left(1 + \frac{\lambda_t \mu}{2}\right) \mathrm{E}_{i_t^{(k)}} \|\boldsymbol{w}_\star - \tilde{\boldsymbol{w}}_{t+1}^{(k)}\|^2 \right) + \lambda_t^2 L^2 + \lambda_t^3 \mu L^2. \tag{18}$$

Taking total expectation of (18) gives

$$2\lambda_t \, \mathrm{E} \left( \frac{1}{n} \sum_{i=1}^n f_i(\boldsymbol{w}_{t+1}) + g(\boldsymbol{w}_{t+1}) - \frac{1}{n} \sum_{i=1}^n f_i(\boldsymbol{w}_\star) - g(\boldsymbol{w}_\star) \right)$$

$$\leq \frac{1}{p} \sum_{k=1}^p \left( \mathrm{E} \|\boldsymbol{w}_\star - \tilde{\boldsymbol{w}}_t^{(k)}\|^2 - \left(1 + \frac{\lambda_t \mu}{2}\right) \mathrm{E} \|\boldsymbol{w}_\star - \tilde{\boldsymbol{w}}_{t+1}^{(k)}\|^2 \right) + \lambda_t^2 L^2 + \lambda_t^3 \mu L^2. \tag{19}$$

Now we separate the cases of $\mu = 0$ and $\mu > 0$:

- If $\mu = 0$, then taking summation of (19) with $t = 0, \ldots, T$ gives

$$2 \sum_{t=0}^T \lambda_t \, \mathrm{E} \left( \frac{1}{n} \sum_{i=1}^n f_i(\boldsymbol{w}_{t+1}) + g(\boldsymbol{w}_{t+1}) - \frac{1}{n} \sum_{i=1}^n f_i(\boldsymbol{w}_\star) - g(\boldsymbol{w}_\star) \right) \leq \frac{1}{p} \sum_{k=1}^p \mathrm{E} \|\boldsymbol{w}_\star - \tilde{\boldsymbol{w}}_0^{(k)}\|^2 + \sum_{t=0}^T \lambda_t^2 L^2.$$

  Since each $\tilde{\boldsymbol{w}}_0^{(k)}$ is initialized at the same point $\tilde{\boldsymbol{w}}_0$, then we have (14) in Theorem 2.7. For a given $T$, if we use constant step sizes $\lambda_t = \|\tilde{\boldsymbol{w}}_0 - \boldsymbol{w}_\star\| / L\sqrt{T}$, then

$$\mathrm{E} \left[ \frac{1}{n} \sum_{i=1}^n f_i(\hat{\boldsymbol{w}}) + g(\hat{\boldsymbol{w}}) \right] - \left( \frac{1}{n} \sum_{i=1}^n f_i(\boldsymbol{w}_\star) + g(\boldsymbol{w}_\star) \right) \leq \frac{\|\tilde{\boldsymbol{w}}_0 - \boldsymbol{w}_\star\| L}{\sqrt{T}},$$

  which shows the $\mathcal{O}(1/\sqrt{t})$ convergence rate in expectation.

- If $\mu > 0$, then plugging in the step size $\lambda_t = \beta/t$ into (19) gives

$$\mathrm{E}\left[\frac{1}{n}\sum_{i=1}^{n} f_i(\boldsymbol{w}_{t+1}) + g(\boldsymbol{w}_{t+1}) - \frac{1}{n}\sum_{i=1}^{n} f_i(\boldsymbol{w}_\star) - g(\boldsymbol{w}_\star)\right]$$

$$\leq \frac{t}{2\beta p}\sum_{k=1}^{p}\mathrm{E}\,\|\tilde{\boldsymbol{w}}_t^{(k)} - \boldsymbol{w}_\star\|^2 - \frac{2t+\mu\beta}{4\beta p}\sum_{k=1}^{p}\mathrm{E}\,\|\tilde{\boldsymbol{w}}_{t+1}^{(k)} - -\boldsymbol{w}_\star\|^2 + \frac{L^2\beta}{2t} + \frac{\beta^2\mu L^2}{2t^2}$$

$$\leq \frac{t}{2\beta p}\sum_{k=1}^{p}\mathrm{E}\,\|\tilde{\boldsymbol{w}}_t^{(k)} - \boldsymbol{w}_\star\|^2 - \frac{t+2}{2\beta p}\sum_{k=1}^{p}\mathrm{E}\,\|\tilde{\boldsymbol{w}}_{t+1}^{(k)} - \boldsymbol{w}_\star\|^2 + \frac{L^2\beta(1+\beta\mu)}{2t},$$

where the last inequality stems from $\mu\beta > 4$ and $1/t^2 < 1/t$ for $t \geq 1$. Multiplying both sides by $t+1$ yields

$$(t+1)\,\mathrm{E}\left[\frac{1}{n}\sum_{i=1}^{n} f_i(\boldsymbol{w}_{t+1}) + g(\boldsymbol{w}_{t+1}) - \frac{1}{n}\sum_{i=1}^{n} f_i(\boldsymbol{w}_\star) - g(\boldsymbol{w}_\star)\right]$$

$$\leq \frac{t(t+1)}{2\beta p}\sum_{k=1}^{p}\mathrm{E}\,\|\tilde{\boldsymbol{w}}_t^{(k)} - \boldsymbol{w}_\star\|^2 - \frac{(t+1)(t+2)}{2\beta p}\sum_{k=1}^{p}\mathrm{E}\,\|\tilde{\boldsymbol{w}}_{t+1}^{(k)} - \boldsymbol{w}_\star\|^2 + \frac{L^2\beta(1+\beta\mu)(t+1)}{2t}$$

$$\leq \frac{t(t+1)}{2\beta p}\sum_{k=1}^{p}\mathrm{E}\,\|\tilde{\boldsymbol{w}}_t^{(k)} - \boldsymbol{w}_\star\|^2 - \frac{(t+1)(t+2)}{2\beta p}\sum_{k=1}^{p}\mathrm{E}\,\|\tilde{\boldsymbol{w}}_{t+1}^{(k)} - \boldsymbol{w}_\star\|^2 + L^2\beta(1+\beta\mu),$$

where, again, the last inequality is due to $(t+1)/2t < 1$ for $t \geq 1$. Now let $t = 1, 2, \ldots, T$ and sum over all inequalities, we get

$$\sum_{t=1}^{T} t\,\mathrm{E}\left[\frac{1}{n}\sum_{i=1}^{n} f_i(\boldsymbol{w}_t) + g(\boldsymbol{w}_t) - \frac{1}{n}\sum_{i=1}^{n} f_i(\boldsymbol{w}_\star) - g(\boldsymbol{w}_\star)\right]$$

$$\leq \frac{1}{\beta p}\sum_{k=1}^{p}\mathrm{E}\,\|\tilde{\boldsymbol{w}}_0^{(k)} - \boldsymbol{w}_\star\|^2 - \frac{T(T+1)}{2\beta p}\sum_{k=1}^{p}\mathrm{E}\,\|\tilde{\boldsymbol{w}}_T^{(k)} - \boldsymbol{w}_\star\|^2 + TL^2\beta(1+\beta\mu)$$

$$\leq \frac{1}{\beta p}\sum_{k=1}^{p}\mathrm{E}\,\|\tilde{\boldsymbol{w}}_0^{(k)} - \boldsymbol{w}_\star\|^2 + TL^2\beta(1+\beta\mu).$$

Since each $\tilde{\boldsymbol{w}}_0^{(k)}$ is initialized at the same point $\tilde{\boldsymbol{w}}_0$, then lowerbounding the left-hand-side by Jensen's inequality and dividing both sides by $1 + 2 + \cdots + T = T(T+1)/2$ yields (15).

$\square$

## 3   Efficient Computation of the Proximal Operators

In this section, we consider how to efficiently implement SDRS for regularized ERM problems (1). In each iteration of Algorithm 1, it involves computing two proximal operators, one for the regularization function $g$ and one for the loss of one sample $f_i$. It is well-known that most regularization functions admit a proximal operator that is efficient to compute, such as the well-known soft-thresholding for $L_1$ norm regularization to promote sparse solutions, or block soft-thresholding for sum of Euclidean norms to promote group sparsity. More comprehensive surveys of similar proximal operators have been studied in many works, for example by Parikh & Boyd (2014). As for the proximal operator for $f_i$ in line 4 of Algorithm 1, it may seem just as challenging as solving the batch problem. However, we will see that there are some interesting properties when $f_i$ involves the loss evaluation of just one or a few data points in a data fitting scenario.

In an ERM formulation for supervised learning, we can often write the one-sample loss function as $f_i(\boldsymbol{w}) = \ell(\boldsymbol{x}_i^\top \boldsymbol{w} - y_i)$ for regression or $f_i(\boldsymbol{w}) = \ell(y_i \boldsymbol{x}_i^\top \boldsymbol{w})$ for binary classification, which means it is a scalar function composed with $\boldsymbol{x}_i^\top \boldsymbol{w}$. Suppose it is differentiable, then by applying the chain rule, we have $\nabla f_i(\boldsymbol{w}) = \boldsymbol{x}_i \ell'(\boldsymbol{x}_i^\top \boldsymbol{w})$,

where $\ell'$ is the derivative of the scalar function $\ell(\cdot)$. This is also true when $f_i$ is nonsmooth that any subgradient is some scaled version of $\boldsymbol{x}_i$. Plugging it into the optimality condition, we get that

$$\mathrm{Prox}_{\lambda f_i}(\boldsymbol{w}) = \boldsymbol{w} - \alpha \boldsymbol{x}_i, \tag{20}$$

for some scalar $\alpha$. The proximal update further reduces to finding the scalar $\alpha$ that solves the following single-variate convex problem,

$$\underset{\alpha}{\text{minimize}} \quad \ell(\boldsymbol{x}_i^\top \boldsymbol{w} - \alpha \|\boldsymbol{x}_i\|^2) + \frac{\alpha^2}{2\lambda_t} \|\boldsymbol{x}_i\|^2. \tag{21}$$

This can be done by simply setting the derivative equal to zero. Due to convexity, its derivative is a monotonic function, so one naive way to find the root is via bisection. In some cases the choice of $\alpha$ even has a closed-form expression. In the supplementary we include the specific proximal operator for the least squares loss, logistic loss, hinge loss, and absolute error loss, which are some of the most widely used losses in ERM.

The special form of the proximal update (20) when the one-sample loss is the composition of a scalar convex function and linear function also gives an interesting explicit form of SDRS. If line 4 of Algorithm 1 is in the form of (20), then line 4 simplifies to $\tilde{\boldsymbol{w}}_{t+1} \leftarrow \boldsymbol{w}_{t+1} - \alpha \boldsymbol{x}_i$; plugging it into line 2 further simplifies SDRS to

$$\boldsymbol{w}_{t+1} \leftarrow \mathrm{Prox}_{\lambda_t g}(\boldsymbol{w}_t - \alpha \boldsymbol{x}_i),$$

which looks surprisingly similar to the proximal stochastic gradient descent, except that the step size $\alpha$ is chosen more delicately via solving (21). As we will see very soon in the next section, this delicate choice often makes a big difference in terms of convergence in practice. Similar arguments can be made for the mini-batch version, making the iterates having the form

$$\boldsymbol{w}_{t+1} \leftarrow \mathrm{Prox}_{\lambda_t g}\left(\boldsymbol{w}_t - \frac{1}{p}\sum_{k=1}^p \alpha^{(k)} \boldsymbol{x}_{i_t^{(k)}}^{(k)}\right),$$

where each $\alpha^{(k)}$ is calculated individually according to its own (21). This makes mini-batch SDRS more sophisticated than mini-batch SGD, as the search direction takes a *weighted* sum of each samples to properly reflect their contribution in the minimization.

All of the aforementioned discussion on the specific implementation of SDRS only applies when the loss function is in the form $f_i(\boldsymbol{w}) = \ell(\boldsymbol{x}_i^\top \boldsymbol{w} - y_i)$ or $\ell(y_i \boldsymbol{x}_i^\top \boldsymbol{w})$. However, even for generic nonlinear programming problems, it is still possible to efficiently evaluate the proximal update. If acquiring first-order information of $f_i$ is not computationally expensive, one can use methods such as L-BFGS (Nocedal & Wright, 2006) or accelerated gradient descent (Nesterov, 1983) to compute $\mathrm{Prox}_{\lambda f_i}(\cdot)$ with approximately $\mathcal{O}(d)$ complexity.

## 4 Experiments

In this section, we show the performance of SDRS with proposed efficient implementation on both classification and regression tasks, and compare it with some widely used alternatives such as SGD-momentum (Liu et al., 2020), Adam (Kingma & Ba, 2014), and AdaBelief (Zhuang et al., 2020). All formulations include an $L_1$ norm regularization, and each SGD-type algorithm is followed by soft-thresholding to handle it. More real data experiments can be found in the appendix.

### 4.1 Classification

We perform binary classification by using SVM and logistic regression with $L_1$ norm regularization on the bank note authentication data set(Dua & Graff, 2017) This includes images of genuine and counterfeit banknotes. There are 1372 images in total. There are five attributes in each image, out of which four are features and one is the target attribute. The target attribute contains 0 and 1, where 0 indicates genuine notes and 1 indicates fake notes. The ratio between the two classes is 55/45 (genuine/counterfeit). We present Figures 1 and 2, illustrating SDRS's performance in loss-time and accuracy-time respectively. Figure 1 illustrates how

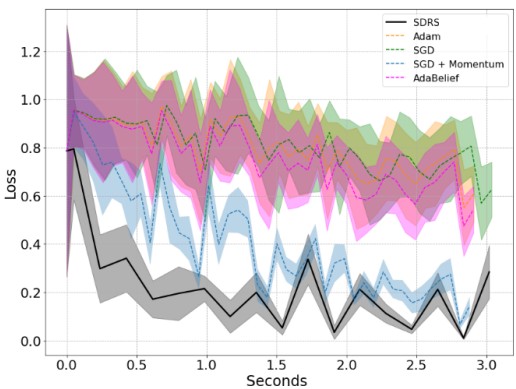

Figure 1: SVM with $L_1$ regularization on Bank Note Authentication: loss per seconds, batch size 1

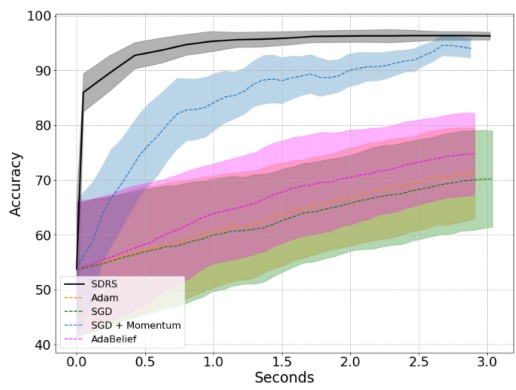

Figure 2: SVM with $L_1$ regularization on Bank Note Authentication: classification accuracy per seconds, batch size 1

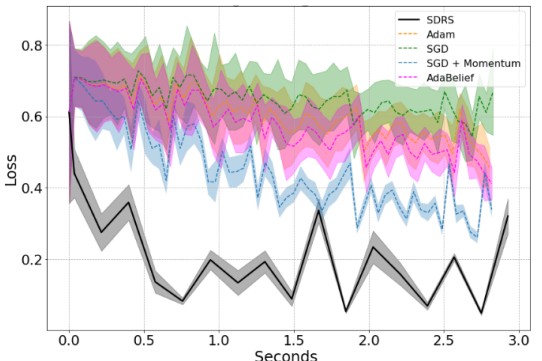

Figure 3: Logistic regression with $L_1$ regularization on Bank Note Authentication: loss per seconds

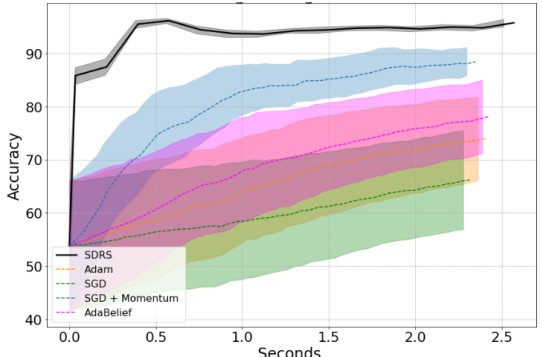

Figure 4: Logistic regression with $L_1$ regularization on Bank Note Authentication: classification accuracy per seconds

the regularized hinge loss value decreases with time (seconds) by comparing different algorithms. Figures 1 and 2 demonstrate that using our efficient implementation, SDRS takes less time to achieve a smaller loss and greater accuracy despite its complexity per iteration.

In Figures 3 and 4 we present results on a logistic regression model with $L_1$ regularization on Bank Note Authentication dataset, representing loss/time and accuracy/time respectively. As seen in Figure 3, SDRS achieves a lower loss value in less time than all other algorithms; the closest performance is achieved by Stochastic Gradient with Momentum. With reference to Figure 4, SDRS achieves the highest accuracy rate in the shortest amount of time, and it has the lowest variance of all of the algorithms.

## 4.2 Classification using mini batch

We also run the classification experiments using the Bank Note Authentication data set. Figures 5 and 6 show the loss and accuracy values of SDRS on Bank Note Authentication dataset for SVM approach. Our modification allows SDRS to handle larger batches. We observe that the SDRS still outperforms all the other methods. We also use Logistic Regression on the same Bank Note Authentication dataset. The Figure 7 and 8 represent the loss and accuracy values respectively. We observe that SDRS reaches a smaller value in the first few iterations.

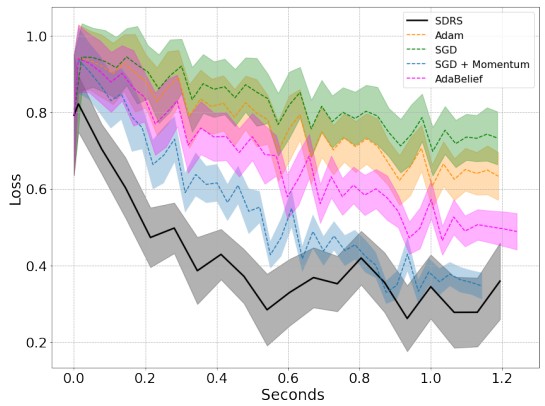

Figure 5: SVM on Bank Note Authentication data set: loss per seconds, batch size 4

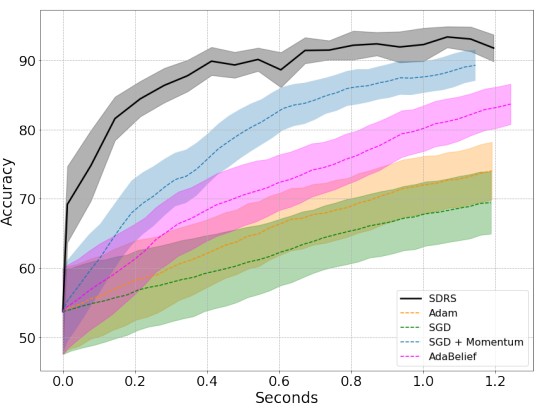

Figure 6: SVM on Bank Note Authentication data set: accuracy per seconds, batch size 4

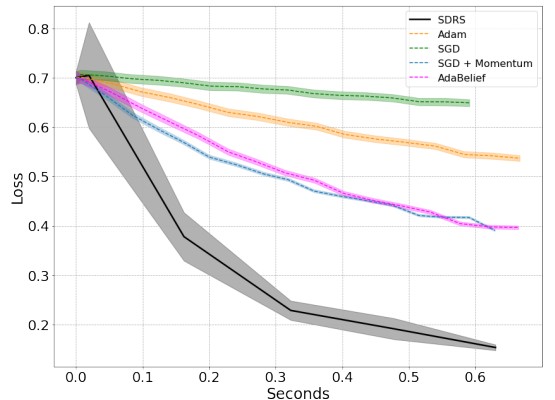

Figure 7: Logistic Regression on Bank Note Authentication data set: loss per seconds, batch size 16

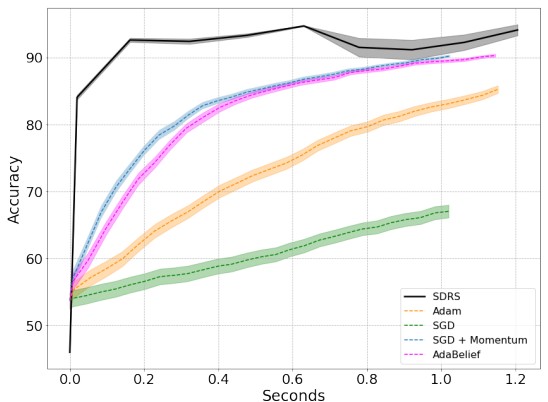

Figure 8: Logistic Regression on Bank Note Authentication data set: accuracy per seconds, batch size 16

### 4.3 Neural network training using mini-batch SPPA

As suggested in §1.2, the proposed mini-batch SDRS in Algorithm 2 also suggests a fully parallelizable mini-batch SPPA when the regularization term $g$ does not exist, as opposed to the one that cannot be fully parallelized by Chadha et al. (2022). In this section we apply mini-batch SPPA to the nonconvex problem of neural network training. We apply it on two famous image classification data sets, CIFAR10 and MNIST, with two different network architectures, CNN and ResNet.

CIFAR10 (Krizhevsky et al., 2010) consists of 50,000 training and 10,000 test images of size $32 \times 32$ in 10 classes. The performance of CNN training on CIFAR10 is shown in Figures 9 and 10. The network consists of 5 layers, each of the first 2 being a combination of $5 \times 5$ convolutional filters and $2 \times 2$ max pooling, and the last three being fully connected. The loss function is cross-entropy loss, and the activation function is ReLU. The batch size is 4. To show the behavior of the algorithms more in detail, we recorded the loss values at every 500 stochastic updates, the experiment is run for five epochs, and in total, 125 updates are presented ($25 \times 500 \times 4$ giving the number of training samples). The accuracy is calculated on the test data at every 500 stochastic update overall for five epochs, using the accuracy calculation in the PyTorch tutorial.

## 5 Conclusion

We studied stochastic Douglas-Rachford splitting (SDRS) in this paper. Our first contribution is to provide convergence analysis of SDRS, thus closing the theoretical gap. We also discussed various implementation

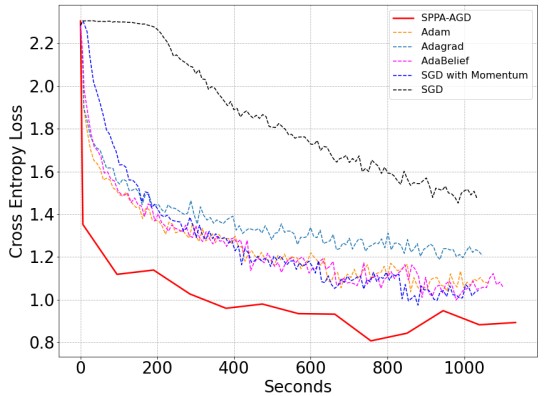

Figure 9: CNN on CIFAR10: cross entropy loss on test data

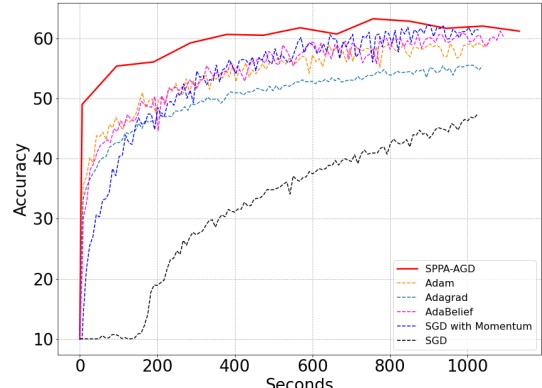

Figure 10: CNN on CIFAR10: prediction accuracy on test data

issues to make it more applicable for practical use of training regularized ERM problems, including a mini-batch strategy that keeps the simplicity of pertinent proximal operators. Our experiments showed that SDRS is able to perform better than many well-known stochastic algorithms for various data sets.

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

## A    Efficient Implementation

In this section, we introduce several efficient methods to calculate the proximal operator update in Algorithm 1 line 5. At first glance, it may seem as hard as solving the batch problem itself, but we will see that there are some interesting properties when $f_i$ involves the loss evaluation of only one or a few data points in a data fitting scenario. On the other hand, it is not clear how to parallelize SPPA with mini-batches, which leaves room for future work.

### A.1   Least squares loss

For the least squares loss $f_i = (y_i - \boldsymbol{x}_i^\top \boldsymbol{w})^2$, the proximal update is a linear least squares problem with closed-form solution $\mathrm{Prox}_{f_i}(\boldsymbol{w}) = (\boldsymbol{I} + \lambda_t \boldsymbol{x}_i \boldsymbol{x}_i^\top)^{-1}(\boldsymbol{w} + \boldsymbol{x}_i y_i)$. It can be efficiently calculated by invoking the Sherman-Morrison formula and avoid directly inverting a $d \times d$ matrix for $\boldsymbol{w} \in \mathbb{R}^d$ as

$$\mathrm{Prox}_{f_i}(\boldsymbol{w}) = \boldsymbol{w} - \frac{\boldsymbol{x}_i^\top \boldsymbol{w} - y_i}{\|\boldsymbol{x}_i\|^2 + 1/\lambda_t} \boldsymbol{x}_i.$$

The update rule looks like, but is not exactly the same as, the normalized least mean squares (NLMS) algorithm (Haykin, 2002). It has appeared in (Pătraşcu, 2020), but we include it here for completeness.

### A.2   Logistic (cross-entropy) loss

Consider the logistic loss $f_i(\boldsymbol{w}) = \log(1 + \exp(-y_i \boldsymbol{x}_i^T \boldsymbol{w}))$ with $y_i = \pm 1$. According to the arguments made in §3, we need to solve the nonlinear equation

$$\alpha = y_i \lambda_t \frac{\exp(\boldsymbol{x}_i^\top \boldsymbol{w} - \alpha \|\boldsymbol{x}_i\|^2)}{1 + \exp(\boldsymbol{x}_i^\top \boldsymbol{w} - \alpha \|\boldsymbol{x}_i\|^2)}.$$

Notice that the right-hand-side is a number between 0 and $\pm \lambda_t$, which gives the initial upper and lowerbound on $\alpha$. Furthermore, we see that $\boldsymbol{x}_i^\top \boldsymbol{w}$ and $\|\boldsymbol{x}_i\|^2$ only need to be calculated once, with $\mathcal{O}(d)$ flops for $\boldsymbol{w} \in \mathbb{R}^d$; each bisection step takes constant time and it needs no more than $20 \sim 30$ scalar computations to render an accurate-enough solution.

### A.3   Hinge loss

Regarding nonsmooth optimization problems, it turns out many of the widely used loss functions admit closed form updates. The main idea is to consider the subgradient calculus, and find the point where 0 is in the subdifferential. Take support vector machine (SVM) as an example, in which the loss function is the hinge loss $f_i(\boldsymbol{w}) = [1 - y_i \boldsymbol{x}_i^\top \boldsymbol{w}]_+$. Its subdifferential is

$$\partial f_i(\boldsymbol{w}) = \begin{cases} \{0\} & 1 - y_i \boldsymbol{x}_i^\top \boldsymbol{w} < 0, \\ \{-y_i \boldsymbol{x}_i\} & 1 - y_i \boldsymbol{x}_i^\top \boldsymbol{w} > 0, \\ \{-\alpha y_i \boldsymbol{x}_i | 0 \le \alpha \le 1\} & 1 - y_i \boldsymbol{x}_i^\top \boldsymbol{w} = 0. \end{cases}$$

Added with the gradient of the proximal term and letting 0 to be an element of the subdifferential set, we get the update rule

$$\mathrm{Prox}_{f_i}(\boldsymbol{w}) = \begin{cases} \boldsymbol{w} & y_i \boldsymbol{x}_i^\top \boldsymbol{w} > 1, \\ \boldsymbol{w} + \lambda_t y_i \boldsymbol{x}_i & y_i \boldsymbol{x}_i^\top \boldsymbol{w}_t < 1 - \lambda_t \|\boldsymbol{x}_i\|^2, \\ \boldsymbol{w} + \lambda_t y_i \boldsymbol{x}_i \frac{1 - y_i \boldsymbol{x}_i^\top \boldsymbol{w}}{y_i \|\boldsymbol{x}_i\|^2} & \text{otherwise.} \end{cases}$$

An interesting observation here is that it looks like the perceptron algorithm with an adaptively chosen step-size.

### A.4   Absolute error loss

Robust regression using absolute error loss $f_i(\boldsymbol{w}) = |y_i - \boldsymbol{x}_i^\top \boldsymbol{w}|$, is another example to nonsmooth convex loss function. The derivation is similar to that of SVM: the subdifferential for the one-sample loss is

$$\partial f_i(\boldsymbol{w}) = \begin{cases} \{\boldsymbol{x}_i\} & y_i - \boldsymbol{x}_i^\top \boldsymbol{w} < 0, \\ \{-\boldsymbol{x}_i\} & y_i - \boldsymbol{x}_i^\top \boldsymbol{w} > 0, \\ \{\alpha \boldsymbol{x}_i | -1 \le \alpha \le 1\} & y_i - \boldsymbol{x}_i^\top \boldsymbol{w} = 0. \end{cases}$$

After adding the proximal term, the update rule is

$$\text{Prox}_{\lambda f_i}(\boldsymbol{w}) = \begin{cases} \boldsymbol{w} - \lambda \boldsymbol{x}_i & y_i - \boldsymbol{x}_i^\top \boldsymbol{w} < -\lambda \|\boldsymbol{x}_i\|^2, \\ \boldsymbol{w} + \lambda \boldsymbol{x}_i & y_i - \boldsymbol{x}_i^\top \boldsymbol{w} > \lambda \|\boldsymbol{x}_i\|^2, \\ \boldsymbol{w} + \frac{y_i - \boldsymbol{x}_i^\top \boldsymbol{w}}{\|\boldsymbol{x}_i\|^2} \boldsymbol{x}_i & \text{otherwise.} \end{cases}$$

### A.5 Generic losses

Even for generic nonlinear programming problems, it is still possible to efficiently evaluate the proximal update beyond merely a simple gradient step. The idea is to apply the limited-memory BFGS algorithm (Nocedal & Wright, 2006), or L-BFGS for short. L-BFGS is a memory-efficient implementation of the famous quasi-Newton algorithm BFGS. In a nut shell, L-BFGS is an iterative algorithm that makes use of the second-order information from the optimization loss function, but does not require solving matrix inverses and only requires explicitly evaluating first-order gradients. For a prescribed number of iterations, it requires one matrix-vector multiplication and multiple vector multiplications. As a result, the overall complexity is again $\mathcal{O}(d)$ if the initial guess of the Hessian matrix is diagonal.

While there are certain limitations for applying L-BFGS to general nonlinear programming problems, we reckon that it fits perfectly in the context of SPPA implementations.

- L-BFGS has to specify a good initial guess of the Hessian approximation matrix. While in many cases people simply use the identity matrix to start, it may result in very poor approximation. Fortunately, thanks to the proximal term, the identity matrix is in fact a very good initial guess for the Hessian matrix for SPPA updates.

- In order to save memory consumption, L-BFGS has to prescribe the number of iterations before running the algorithm. Obviously, if the prescribed number of iteration is too large, we incur unnecessary computations, while if it is too small we need to invoke another round with a new estimated Hessian matrix. However, again in the context of SPPA, the proximal term naturally provides a good initialization $\boldsymbol{w}_t$. Our experience show that prescribing 10 iterations of L-BFGS updates is more than enough to obtain accurate solutions.

On the other hand, it perhaps makes more sense to simply use some more advanced first-order methods such as Nesterov's accelerated gradient descent (1983) to calculate the proximal update. Both L-BFGS and accelerated gradient descent evaluates the gradient of the loss function with $\mathcal{O}(d)$ complexity, and the question is how to leverage convergence rate versus sophistication.

## B  Miscellaneous proofs

### B.1  Any diminishing step size rule guarantees expected convergence

It is our observation that a lot of the convergence analysis using diminishing step sizes requires that the step sizes are square summable. Take equation (2) as an example, on the right-hand-side, the second term in the numerator is typically assumed to be finite, while the denominator goes to infinity, therefore the entire right hand side goes to zero. This requirement would rule out step size rules such as $\lambda_t = \beta/\sqrt{t}$ since it is not square summable. The purpose of this subsection is to show that the common square summable requirement is not in fact necessary as the quotient still goes to zero. We are not claiming to be the first to notice this, so it is only included for completeness.

**Proposition B.1.** *For a nonnegative sequence $\lambda_1, \lambda_2, \ldots$, that satisfies $\lambda_t \to 0$ and $\sum_{t=1}^{\infty} \lambda_t = \infty$, we have that*

$$\frac{\sum_{t=1}^{\infty} \lambda_t^2}{\sum_{t=1}^{\infty} \lambda_t} = 0.$$

*Proof.* Let $\epsilon > 0$, since $\lambda_t \to 0$, there exists an integer $T_1$ such that $\lambda_t < \epsilon$ for all $t > T_1$. There also exists an integer $T_2$ such that

$$\sum_{t=1}^{T_2} \lambda_t \geq \frac{1}{\epsilon} \sum_{t=1}^{T_1} \lambda_t^2,$$

since the right-hand-side is a fixed number and $\sum_{t=1}^{\infty} \lambda_t = \infty$. Then for any $T > \max(T_1, T_2)$, we have

$$\frac{\sum_{t=1}^{T} \lambda_t^2}{\sum_{t=1}^{T} \lambda_t} = \frac{\sum_{t=1}^{T_1} \lambda_t^2}{\sum_{t=1}^{T_2} \lambda_t + \sum_{t=T_2+1}^{T} \lambda_t} + \frac{\sum_{t=T_1+1}^{T} \lambda_t^2}{\sum_{t=1}^{T_1} \lambda_t + \sum_{t=T_1+1}^{T} \lambda_t}$$

$$\leq \frac{\sum_{t=1}^{T_1} \lambda_t^2}{\sum_{t=1}^{T_2} \lambda_t} + \frac{\sum_{t=T_1+1}^{T} \lambda_t^2}{\sum_{t=T_1+1}^{T} \lambda_t} \leq \epsilon + \frac{\epsilon \sum_{t=T_1+1}^{T} \lambda_t}{\sum_{t=T_1+1}^{T} \lambda_t} \leq 2\epsilon.$$

Since the inequality holds for any $\epsilon > 0$, we can let $\epsilon$ be arbitrarily close to zero, and there will be $T$ large enough to make the inequality hold. This proves the proposition. $\square$

## B.2 Convergence in probability

In this convergence analysis, we will use the following well-known theorem (Bertsekas, 2011).

**Theorem B.2** (Supermartingale Convergence Theorem). *Let $X_t$, $Y_t$, and $Z_t$, $t = 0, 1, \ldots$, be three sequences of random variables and let $\mathcal{F}_t, t = 0, 1, \ldots$, be sets of random variables such that $\mathcal{F}_t \subset \mathcal{F}_{t+1}$ for all $t$. Suppose that:*

1. *The random variables $X_t$, $Y_t$, and $Z_t$ are nonnegative, and are functions of the random variables in $\mathcal{F}_t$.*

2. *For each $t$, we have*

$$\mathrm{E}[X_{t+1}|\mathcal{F}_t] \leq X_t - Y_t + Z_t.$$

3. *There holds, with probability 1, $\sum_{t=0}^{\infty} Z_t \leq \infty$.*

*Then we have $\sum_{t=0}^{\infty} Y_t \leq \infty$, and the sequence $X_t$ converges to a nonnegative random variable $X$, with probability 1.*

**Corollary B.3** (Convergence in probability with constant step sizes). *Suppose all $f_1, \ldots, f_n$ and $g$ are convex, and Assumption 2.1 holds. If a solution $\boldsymbol{w}_\star$ exists, then with initialization $\tilde{\boldsymbol{w}}_0$, with probability 1, the sequence $\boldsymbol{w}_1, \ldots, \boldsymbol{w}_T$ generated by SDRS (Algorithm 1) or mini-batch SDRS (Algorithm 2) with a constant step size $\lambda_t = \lambda$ satisfies*

$$\inf_t \left[ \frac{1}{n} \sum_{i=1}^{n} f_i(\boldsymbol{w}_t) + g(\boldsymbol{w}_t) \right] < \frac{1}{n} \sum_{i=1}^{n} f_i(\boldsymbol{w}_\star) + g(\boldsymbol{w}_\star) + \frac{\lambda L^2}{2}.$$

*Proof.* Rewriting (8) as

$$\mathrm{E}_{i_t} \|\tilde{\boldsymbol{w}}_{t+1} - \boldsymbol{w}_\star\|^2 \leq \|\tilde{\boldsymbol{w}}_t - \boldsymbol{w}_\star\|^2 - 2\lambda \left( \frac{1}{n} \sum_{i=1}^{n} f_i(\boldsymbol{w}_{t+1}) + g(\boldsymbol{w}_{t+1}) - \frac{1}{n} \sum_{i=1}^{n} f_i(\boldsymbol{w}_\star) - g(\boldsymbol{w}_\star) \right) + \lambda^2 L^2.$$

Let $X_t = \|\tilde{\boldsymbol{w}}_t - \boldsymbol{w}_\star\|^2$, $Z_t = 0$, and

$$Y_t = 2\lambda \left( \frac{1}{n} \sum_{i=1}^{n} f_i(\boldsymbol{w}_{t+1}) + g(\boldsymbol{w}_{t+1}) - \frac{1}{n} \sum_{i=1}^{n} f_i(\boldsymbol{w}_\star) - g(\boldsymbol{w}_\star) \right) - \lambda^2 L^2,$$

as well as filtration $\mathcal{F}_t = \{\boldsymbol{w}_{t+1}, \ldots, \boldsymbol{w}_0, \tilde{\boldsymbol{w}}_t, \ldots, \tilde{\boldsymbol{w}}_0\}$, which would imply $\mathcal{F}_t \subset \mathcal{F}_{t+1}$, then according to the Supermartingale Convergence Theorem B.2,

$$\sum_{t=1}^{\infty} 2\lambda \left( \frac{1}{n} \sum_{i=1}^{n} f_i(\boldsymbol{w}_{t+1}) + g(\boldsymbol{w}_{t+1}) - \frac{1}{n} \sum_{i=1}^{n} f_i(\boldsymbol{w}_\star) - g(\boldsymbol{w}_\star) - \lambda^2 L^2 \right) < \infty,$$

with probability 1. Divide both sides by $2\lambda$ gives,

$$\sum_{t=1}^{\infty}\left(\frac{1}{n}\sum_{i=1}^{n}f_i(\boldsymbol{w}_{t+1})+g(\boldsymbol{w}_{t+1})-\frac{1}{n}\sum_{i=1}^{n}f_i(\boldsymbol{w}_\star)-g(\boldsymbol{w}_\star)-\frac{\lambda L^2}{2}\right)<\infty.$$

Since this infinite sum is finite, we must have

$$\inf_t\left(\frac{1}{n}\sum_{i=1}^{n}f_i(\boldsymbol{w}_{t+1})+g(\boldsymbol{w}_{t+1})-\frac{1}{n}\sum_{i=1}^{n}f_i(\boldsymbol{w}_\star)-g(\boldsymbol{w}_\star)-\frac{\lambda L^2}{2}\right)\leq 0.$$

This is because if the infimum is strictly positive, say $\epsilon$, then the infinite sum would be bigger than $\epsilon$ times infinite, which contradicts the summable conclusion from the supermartingale convergence theorem. $\qquad\square$

**Corollary B.4** (Convergence in probability with diminishing step sizes)**.** *Suppose all $f_1,\ldots,f_n$ and $g$ are convex, and Assumption 2.1 holds. If a solution $\boldsymbol{w}_\star$ exists, then with initialization $\tilde{\boldsymbol{w}}_0$, with probability 1, the sequence $\boldsymbol{w}_1,\ldots,\boldsymbol{w}_T$ generated by SDRS (Algorithm 1) or mini-batch SDRS (Algorithm 2) with diminishing step sizes such that $\lambda_t\to 0$, $\sum_{t=1}^{\infty}\lambda_t=\infty$, and $\sum_{t=1}^{\infty}\lambda_t^2<\infty$ satisfies*

$$\inf_t\left[\frac{1}{n}\sum_{i=1}^{n}f_i(\boldsymbol{w}_t)+g(\boldsymbol{w}_t)\right]\to\frac{1}{n}\sum_{i=1}^{n}f_i(\boldsymbol{w}_\star)+g(\boldsymbol{w}_\star).$$

*Proof.* Following the first few steps in the proof of Corollary B.3, but replacing $\lambda$ with $\lambda_t$, we have

$$\mathrm{E}_{i_t}\|\tilde{\boldsymbol{w}}_{t+1}-\boldsymbol{w}_\star\|^2\leq\|\tilde{\boldsymbol{w}}_t-\boldsymbol{w}_\star\|^2-2\lambda_t\left(\frac{1}{n}\sum_{i=1}^{n}f_i(\boldsymbol{w}_{t+1})+g(\boldsymbol{w}_{t+1})-\frac{1}{n}\sum_{i=1}^{n}f_i(\boldsymbol{w}_\star)-g(\boldsymbol{w}_\star)\right)+\lambda_t^2 L^2.$$

This time we let $X_t=\|\tilde{\boldsymbol{w}}_t-\boldsymbol{w}_\star\|^2$, $Z_t=\lambda_t^2 L^2$, and

$$Y_t=2\lambda_t\left(\frac{1}{n}\sum_{i=1}^{n}f_i(\boldsymbol{w}_{t+1})+g(\boldsymbol{w}_{t+1})-\frac{1}{n}\sum_{i=1}^{n}f_i(\boldsymbol{w}_\star)-g(\boldsymbol{w}_\star)\right),$$

as well as filtration $\mathcal{F}_t=\{\boldsymbol{w}_{t+1},\ldots,\boldsymbol{w}_0,\tilde{\boldsymbol{w}}_t,\ldots,\tilde{\boldsymbol{w}}_0\}$. Since we assume $\lambda_t$ is square summable, we can invoke the Supermartingale Convergence Theorem B.2 to conclude that

$$\sum_{t=1}^{\infty}2\lambda_t\left(\frac{1}{n}\sum_{i=1}^{n}f_i(\boldsymbol{w}_{t+1})+g(\boldsymbol{w}_{t+1})-\frac{1}{n}\sum_{i=1}^{n}f_i(\boldsymbol{w}_\star)-g(\boldsymbol{w}_\star)\right)<\infty$$

with probability 1. This means

$$\inf_t\left(\frac{1}{n}\sum_{i=1}^{n}f_i(\boldsymbol{w}_{t+1})+g(\boldsymbol{w}_{t+1})-\frac{1}{n}\sum_{i=1}^{n}f_i(\boldsymbol{w}_\star)-g(\boldsymbol{w}_\star)\right)=0;$$

otherwise, if the infimum is $\epsilon>0$, then the infinite sum is bigger than $\epsilon\sum_{t=1}^{\infty}\lambda_t=\infty$, since we assume $\lambda_t$ is not summable, which contradicts the conclusion from the supermartingale convergence theorem. $\qquad\square$

## C  Additional Experiments

### C.1  Classification on IMDB

IMDB is a large movie review data set used to analyze binary sentiment (Maas et al., 2011). We have 25,000 movie reviews for training and 25,000 movie reviews for testing. To conduct our experiment, we used a bag of words format. Positive to negative sentiment ratio is 50/50.

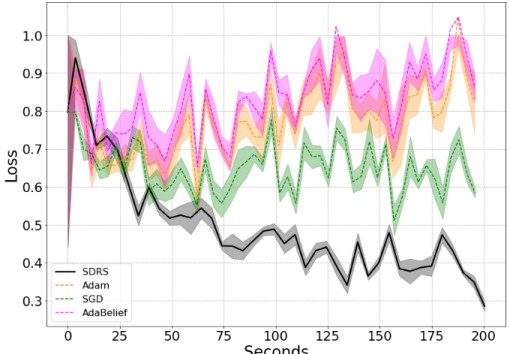

Figure 11: SVM with $L_1$ regularization on IMDB: loss per seconds

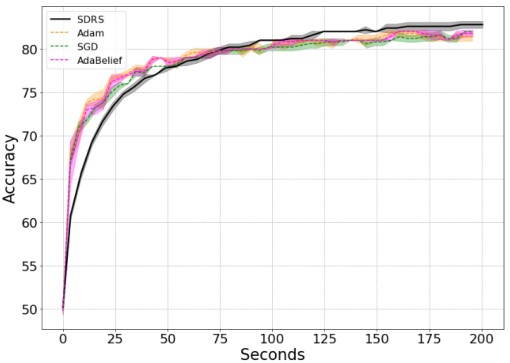

Figure 12: SVM with $L_1$ regularization on IMDB: classification accuracy per seconds

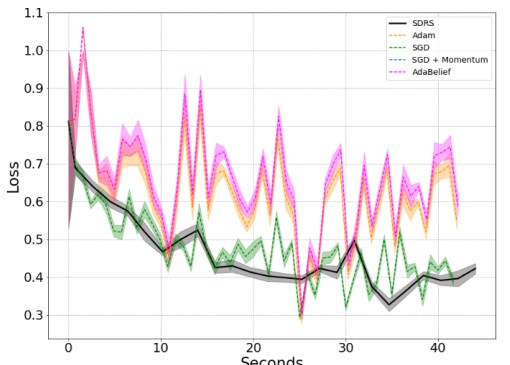

Figure 13: Logistic regression with $L_1$ regularization on IMDB: loss per seconds

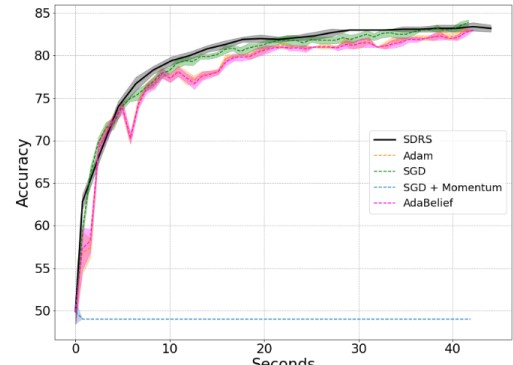

Figure 14: Logistic regression with $L_1$ regularization on IMDB: classification accuracy per seconds

We use SVM and logistic regression for IMDB classification, as we did for Bank Note Authentication. SDRS outperforms the other algorithms when we use logistic regression to classify. Surprisingly, SGD Momentum diverges with this data set in the logistic regression problem. The experiment shows that SGD Momentum is less stable than SDRS, despite performing similarly in most cases. All algorithms are tested using different parameters, and the results presented here are the best results for each algorithm. The logistic loss is shown in Figure 13, and the accuracy per second is shown in Figure 14. Figure 13 shows that logistic loss decreases with SDRS, whereas it does not decrease quite as much with other algorithms. Furthermore, SDRS achieves a higher test accuracy than other algorithms in Figure 14. The performance of SDRS is similar but not notably better than the state-of-the-art methods when we use SVM.

## C.2    Regression

We perform linear regression with $L_1$ regularization and show the results of different algorithms including our efficient implementation on both Bike Sharing and Hotel Average Daily Rates (ADR) datasets (Antonio et al., 2019).

**Bike Sharing.**    Bike Sharing is a publicly available data set containing hourly or daily count of rental bikes as well as environmental and seasonal settings. The data set we are using is from UC Irvine Machine Learning Repository. [1] The core data set consists of two years of historical logs for years 2011 and 2012 from the Capital Bike share system, Washington, DC, USA (Fanaee-T & Gama, 2013). There are 17379 samples in this dataset (12512 training samples, 3476 test samples, 1391 validation samples) with 16 features to predict the hourly rental bike count.

---

[1] https://archive.ics.uci.edu/ml/datasets/bike+sharing+dataset

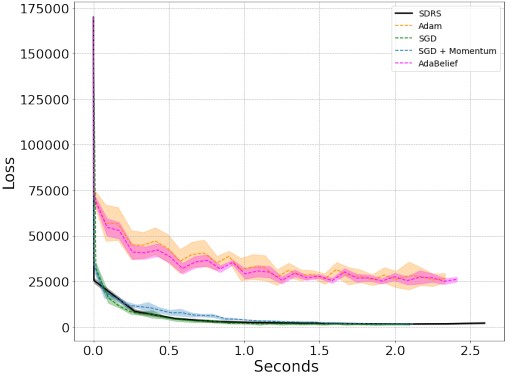

Figure 15: Linear regression with $L_1$ regularization on Bike Sharing: mean squared error loss per seconds.

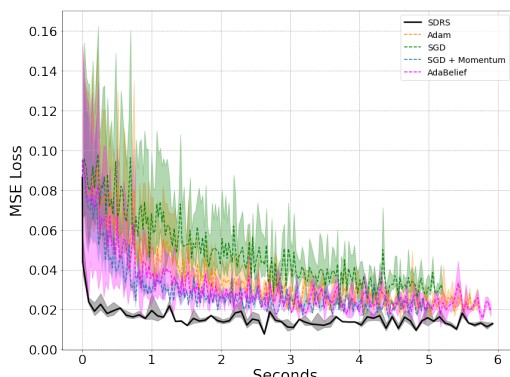

Figure 16: Linear regression with $L_1$ regularization on Hotel ADR: mean squared error loss per seconds.

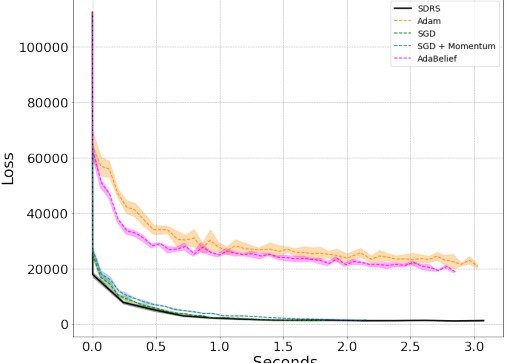

Figure 17: Linear regression on Bike Sharing: loss per second, batch size 4

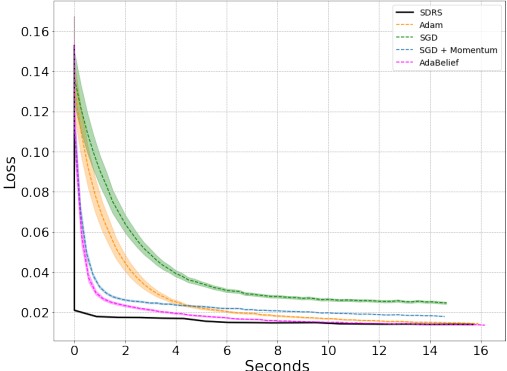

Figure 18: Linear regression on Hotel ADR data set: loss per second, batch size 10

On the bike sharing data set, we perform linear regression with $L_1$ regularization and compare the performance of different optimization algorithms. Figure 15 illustrates the decrease in the loss values for different algorithms over time. The SDRS algorithm outperforms the other algorithms by reaching a smaller value in a shorter amount of time. We tried different settings for the algorithms, and the results presented in Figure 15 show the SDRS learning rate to be $1/\sqrt{t}$, where $t$ is the number of iterations. Similarly, SDRS also offers the benefit of being stable, as demonstrated in Figure 15. Since the dimension is very low for this data set, SDRS is not significantly faster than SGD or SGD momemtum. With mini-batch, as shown in Figure 17, we can see that SDRS takes fewer number of iterations to reach a smaller loss value compared to other methods.

**Hotel Average Daily Rates (ADR).** Hotel ADR is a publicly available data set consisting of hotel demand data for two different types of hotels; resort hotels and city hotels. Each hotel data set consists of 40,060 samples with 31 features each to predict average daily rate values (Antonio et al., 2019). In our experiments, we used resort hotels with 40,060 samples (30045 training, 10015 test) with 8 features.

We perform linear regression with $L_1$ regularization on the Hotel ADR dataset and compare the performances of different optimization algorithms. Figure 16 shows how loss values have decreased over time for multiple algorithms. It takes SDRS less than a second to converge to a very small loss value. We tried different settings for the algorithms, and the results are represented in Figure 16 where the learning rate for SDRS is $1/t$, where $t$ is the number of iterations. With mini-batch, as in Figure 18, we can see the decreasing loss value over iterations. SDRS outperforms all the methods and provides an almost negligible error bar. The performance can be further showcased using parallelization over different batches. Nevertheless, the current results are good enough to celebrate.

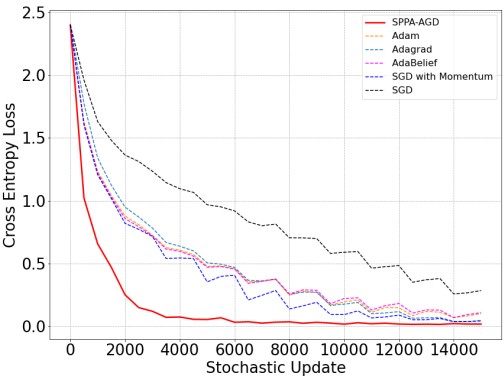

Figure 19: ResNet on CIFAR10: cross entropy loss, batch size 32

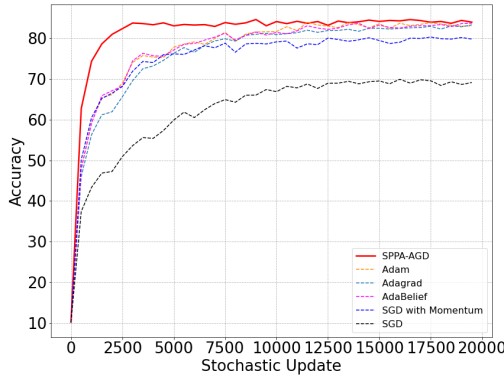

Figure 20: ResNet on CIFAR10: prediction accuracy, batch size 32

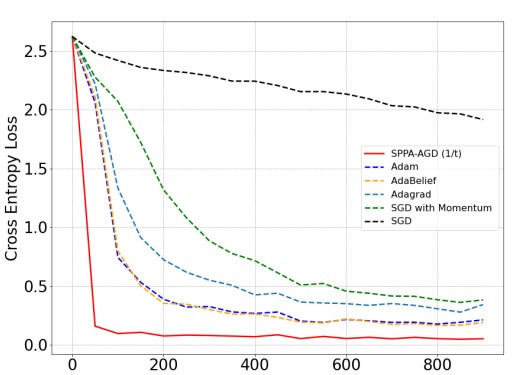

Figure 21: CNN on MNIST: cross entropy loss, batch size 64

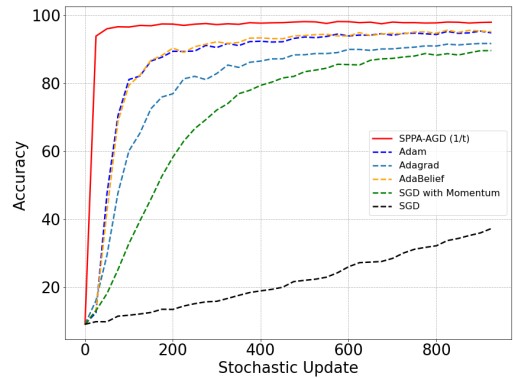

Figure 22: CNN on MNIST: prediction accuracy, batch size 64

### C.3 Additional neural network training

To show that mini-batch SPPA works not only on the CNN architecture, we also tried training a residual neural network (ResNet) on CIFAR10. For the details about the network, one can refer to the 20 layer ResNet architecture on CIFAR10 (He et al., 2016). In our settings, the batch size is 32. The results are shown in Figures 19 and 20. As we can see, the performance is indeed consistent as in the CNN case.

Finally, we show the training performance on the MNIST handwritten digits data set (LeCun et al., 2010). It consists of 60,000 training, 10,000 test images of size $28 \times 28$ in 10 classes. The network we used for this experiment is based on a convolutional deep neural network. It consists of 5 layers, each of the first 3 is a combination of $3 \times 3$ convolutional filters and $2 \times 2$ max pooling with a stride of 2 and the last two fully connected. The loss function is cross-entropy loss, and the activation function is RELU. The batch size is 64. To show the behavior of the algorithms more in detail, we recorded the loss values at every 25 stochastic update, the experiment is run for one epoch, and in total, 38 updates are presented ($25 \times 38 \times 64$ giving the number of training samples). The accuracy is calculated on the test data at every 500 stochastic update, using the accuracy calculation in PyTorch tutorial. [2] As observed in Figures 21 and 22, SPPA-based methods outperform all SGD-based algorithms in terms of both the cross-entropy loss and prediction accuracy on the test set.

---

[2]https://github.com/pytorch/tutorials/blob/master/beginner_source/blitz/cifar10_tutorial.py

