# OpenReview forum: "Stochastic Douglas-Rachford Splitting for Regularized Empirical Risk Minimization: Convergence, Mini-batch, and Implementation"
_TMLR — Accepted by TMLR_

### Review · Reviewer_oBE3 · 2022-10-10

**Summary Of Contributions:**

This paper provides analyses for finite-sum (randomized, incremental and mini-batch) variants of the Douglas-Rachford splitting method. Assuming convexity and Lipschitzness of the individual functions in the finite sum, these methods attain rates $O(1/\sqrt T)$ and $O(1/T)$ for the convex and strongly convex cases (regarding the regularization term) respectively. The paper also provides some numerical experiments, where the proposed algorithms compare favorably to existing alternatives in the literature.

**Requested Changes:**

1. The guarantees of the algorithm are stated for the best iterate, which seems inconvenient, as it would be too expensive to implement. I recommend instead stating the accuracy guarantees for an iterate selected at random (which would lead to entirely analogous upper bounds).
2. In the proof of theorem 2.2 (as well as the other convergence proofs of the paper), there is one step I couldn't understand. This is the equality before eqn. (5): $=\frac{1}{\lambda_t}(\tilde w_{t+1}-\tilde{w}_t)^{\top}(w_{\ast}-\tilde{w}_{t+1})$. In particular, for this equality to hold, the last term of the preceding equation must be zero, which I couldn't follow why it is true. So please add a comment about this step.
3. In the proof of Theorem 2.7, at some point it is said that the rest of the steps follow analogously as in the proof of Theorem 2.2. I would encourage the authors to: (i) Improve the analysis to leverage variance reduction for faster rates, and (ii) add the details of the rest of the proof.

**Strengths And Weaknesses:**

Strenghts:
1. The setting studied in the paper is relevant: implementing stochastic proximal updates is in many cases tractable (as the authors show), and not much work has been devoted to investigating accuracy guarantees for these types of oracles.
2. The analyses in the paper are quite natural, so they can be useful for further work on the topic.
3. Numerical experiments are very encouraging.

Weaknesses:
1. The main weakness of this paper is the lack of quantitative improvements in complexity, compared to more classical results. For example, Theorem 2.2 shows exactly the same convergence rate as that of SGD. In fact, I believe that the stochastic proximal gradient method would also have the same convergence rate under the same assumptions made by the authors, but please correct me if I am wrong. The natural question is then why one would prefer the proposed methods of this paper. Further, mini-batching usually would boost the rates by its variance reduction, but this is not at all reflected in Theorem 2.7. Somehow, while it is nice that the results of the work reproduce existing rates in the literature, shouldn't one expect better theoretical results from these stronger oracles?

---

> ### Author Response · Authors · 2022-10-11
> **Thank you for your positive assessments**
>
> We would like to quickly respond to the major weakness and some requested changes. We will revise the paper accordingly and submit the revision after receiving other reviews.
>
> Major weakness:
>
> - Convergence rate. We were hoping to achieve better results as well, to be honest. The reason we decided to settle with the current result is as follows: As explained in $\S1.1$, SDRS reduces to SPPA when there is no regularization, and the convergence of SPPA has been studied by Bertsekas (2011). What Bertsekas showed, essentially, was $O(1/\sqrt{t})$ convergence of SPPA for convex functions. If we could show faster convergence of SDRS, then as a corollary it would also be a significant improvement to Bertsekas analysis of SPPA. While it is not mission impossible, it is perhaps a good indication that this is the best one can obtain *in theory*. Its fast convergence in practice, even though it may not be theoretically proven, is still a good reason to use it in practice; maybe an analogy can be made about Adam, which has been widely adopted in practice but has not been proven to converge faster than SGD, to the best of our knowledge.
>
> - Mini-batch. As far as we know, mini-batch is not able to theoretically boost the rate for SGD, at least not in expectation. SVRG achieves a faster rate by periodically obtaining *the full gradient*. On the other hand, since we only showed *expected* convergence, our conjecture is that mini-batch does improve the variance of the convergence. However, to the best of our knowledge, there has not been analysis about the variance of the convergence of any stochastic algorithm. Please correct us if we missed something.
>
> Requested changes:
>
> 1. This is a really good suggestion. We will revise accordingly.
>
> 2. Please note that the term on the second last line of (5) is not the same as the first term in line 3. In the third line of (5), the two terms are both vector inner products where one of the vector is $(\tilde{w}_{t+1}-\tilde{w}_t)$. Therefore the two inner-products are combined into one, which gives the next line.
>
> 3. Thanks for the suggestion. We will add the rest of the proof of Theorem 2.7 in the appendix as it really is verbatim to the proof of Theorem 2.2 after eqn. (5).
>
> We would like to express our sincere gratitude again for your positive assessment and the quick review.

---

> > ### Comment · Reviewer_oBE3 · 2022-11-03
> > **After authors rebuttal**
> >
> > Thank you. You are right about your claims (at least, to the best of my knowledge). I have no further questions.

---

### Review · Reviewer_toTG · 2022-10-20

**Summary Of Contributions:**

This paper considers the stochastic Douglas Rachford splitting (SDRS) algorithm to minimize F+g, where F is a sum of n terms f_i. They consider two variants of the algorithm. One vanilla SDRS where a randomly chosen term of the sum is considered per iteration, and one minibatch version which works on a lifted space (p sequences of iterates, where p is the minibatch size).

The main contribution is to establish convergence rates in the convex case and in the case where g is strongly convex. The paper always assumes that f_i is Lipschitz.

**Requested Changes:**

MAJOR:

-What is Algorithm 2 with n=1? It should a known primal dual algorithm applied to the consensus constraints.
-Add a related works section, and discuss the results of concurrent works
-Adjust the rhetoric (the points that I mentioned above).

MINOR:

-Change the title of Section B.1. What is the point of Section B.1?
-Page 2: "Suppose we consider p samples in each stochastic step, not only is the complexity be in general p^2 d + p^3, but [...]" Could you explain this statement please?


**Strengths And Weaknesses:**

Strengths:

-The problem and the contributions are very clear.

-The convergence rates are exactly the ones that one would expect. There is no surprise here. The rates recover the rates of the stochastic proximal point as a special case.

-Section 3 provides an interesting discussion of application of SDRS in ML. In particular, the equation after Eq 15 reminds me the difference between two of the most famous signal processing algorithms: LMS (which actually corresponds to SGD) and normalized LMS (which corresponds to SPPA). However, I think that Section A is covered in www.proximity-operator.net


Weaknesses:

- The paper does not do a good job in the literature review. In particular, there are few comments which are not appropriate for a research paper in my opinion. Examples:
 "This means that the correct way of doing mini-batch for SPPA should be Algorithm 2 with line 2 being the average, which is not the same as any of the proposed mini-batch methods in (Chadha et al., 2022)."
Why would your algorithm be the correct one and the other not being correct? At the stage of the paper, the authors haven't explained what Algorithm 2 is. In particular, does Algorithm 2 corresponds to any known algorithm in the deterministic case? I suspect that, in the deterministic case (i.e, n=1), Algorithm 2 is equivalent to Douglas Rachford applied to a lifted problem. Or maybe it is a known primal dual algorithm that can handle a consensus constraint? By the way, is Algorithm 2 converging with a constant step if n=1?
"The latter two [papers] are either preprint or abridged work without full convergence analysis"
"The latter two" papers seem to be the most related papers to this one, i.e. they are the papers which considered SDRS previously. Therefore, it is not fair to rule them out this way, even more in a section which is not dedicated to related work. In a related work section, one would need to compare the results, this is what I would have expected. Moreover, the way they are ruled out is incorrect: with a small research effort, I found out that Shi & Liu was published in ACML workshop on Learning on big data 2016, and that the full convergence analysis of Salim et al is in the supplementary material available on the website of the author for example.

-Why is Assumption 2.1 (Lipschitzness of all f_i) needed, since this is not needed in the deterministic case? DRS is a proximal method, it should rely on milder assumptions than subgradient methods because we have to compute proximity operators instead of subgradients, which is more computationally demanding. Otherwise, what is the point of using a stochastic proximal method compared to a subgradient? Finally, I don't think that "this [Lipschitz] assumption is so mild that most functions do satisfy it". Because of this assumption, the authors cannot handle strong convexity in the stochastic term F.


-[COMMENT IRRELEVANT FOR PUBLICATION IN TMLR]: The contributions might be of limited interest to the ML community. I think that the theoretical gap closed in this paper (the convergence rates) remained a gap mostly because the community has a limited interest in this problem. Although the analysis is quite standard, it is quite clean, and it is nice to see a paper presenting the convergence rates of SDRS with the correct values.

---

> ### Author Response · Authors · 2022-10-30
> **Thank you for your positive comments**
>
> We will revise the paper accordingly when we receive all the reviews. To respond to some of your valuable comments:
>
> > Why would your algorithm be the correct one and the other not being correct?
>
> This was indeed a valid criticism and we will reword it in the revision to "Algorithm 2 is, in our opinion, the more appropriate mini-batch extension to SDRS". We hold this opinion because we think a valid extension should have both the properties of parallelizable and convergence. Chadha et al. (2022) considers *approximate* proximal updates, and the mini-batch is parallelizable only if the approximation reduces to a stochastic gradient step. Our mini-batch, on the other hand, is both parallelizable and a proper proximal algorithm.
>
> > does Algorithm 2 corresponds to any known algorithm in the deterministic case?
>
> Yes, it corresponds to consensus ADMM if we rearrange the updates of ADMM into the form of Douglas-Rachford. The deterministic case means $n=p$ (not $n=1$). The expected convergence is exactly the same as Algorithm 1 for all values of $p$, including $p=1$, as we have shown in Theorem 2.7.
>
> > "The latter two" papers
>
> Thank you so much for pointing them out. We will carefully reword the description of these papers and put them in the Related Work section in the revision.
>
> > Why is Assumption 2.1 (Lipschitzness of all f_i) needed, since this is not needed in the deterministic case?
>
> This assumption is needed for the analysis of SPPA in (Bertsekas, 2011). Since SDRS reduces to SPPA when $g$ is absent, we would prove a much better convergence for SPPA than (Bertsekas, 2011) if we could drop the Lipschitz continuity assumption of all $f_i$. This is why we think it is a reasonable assumption.
>
> > Otherwise, what is the point of using a stochastic proximal method compared to a subgradient?
>
> First off, the empirical performance is really good. It is not uncommon to see an algorithm working really well in practice, but its theoretical convergence is not superior (e.g., ADMM or Adam). Second, SPPA was not shown to converge faster than SGD by Bertsekas (2011), so perhaps this is indeed the best one can show. Finally, we think it's better to know that it converges no worse than SGD than not knowing it at all.
>
> > I don't think that "this [Lipschitz] assumption is so mild that most functions do satisfy it". Because of this assumption, the authors cannot handle strong convexity in the stochastic term F.
>
> It is true that a function cannot be both Lipschitz continuous and strongly convex. Fortunately, our analysis in $\S2.2$ only assumes strong convexity on $g$ (no Lipschitz continuity on $g$) and Lipschitz continuity on each $f_i$ (no strongly convexity on any $f_i$), so all analysis on the strongly convex case still holds.
>
> > What is the point of Section B.1?
>
> It is our observation that a lot of the convergence analysis using diminishing step sizes requires that the step sizes are square summable. Take equation (2) in our manuscript as an example, on the right-hand-side, the second term in the numerator is typically assumed to be finite, while the denominator goes to infinity, therefore the entire right hand side goes to zero. This requirement would rule out step size rules such as $1/\sqrt{t}$ since it's not square summable. The purpose of $\S$B.1 is to show that the common square summable requirement is not in fact necessary as the quotient still goes to zero. We don't claim to be the first to notice this, so we are not claiming it as a major contribution. However, we do not see it being mentioned in the literature very often, so we feel the need to put it in the supplementary. If the reviewer thinks this is common knowledge, then we will remove this section in the revision.
>
> > "Suppose we consider $p$ samples in each stochastic step, not only is the complexity be in general $p^2d + p^3$, ..." Could you explain this statement please?
>
> If each stochastic step involves $p$ samples, a naive implementation would minimize $(1/p)\sum_i f_i(\theta) + (1/2\lambda)||\theta-\theta^{(t)}||^2$. The summation of $p$ terms means that none of the efficient implementations mentioned in $\S$A could be applied. If the minimization is computed using a generic Newton's method / interior point method, the complexity would be $O(p^2d+p^3)$; if $p<d$, one can still implement it so that the complexity is cubic in the smaller dimension, not the bigger dimension; yet this is still too high and not parallelizable.
>
> > [COMMENT IRRELEVANT FOR PUBLICATION IN TMLR]
>
> We respectfully disagree on this comment. On the front page of the TMLR website, it is written: "TMLR emphasizes technical correctness over subjective significance, to ensure that we facilitate scientific discourse on topics that are deemed less significant by contemporaries but may be important in the future." Regardless of whether the theoretical gap is due to the lack of interests in the community, we think the contribution of this work fits the mission of TMLR well.

---

> > ### Comment · Reviewer_toTG · 2022-11-14
> > **Thanks for the answers**
> >
> > Thank you for answering my concerns. I am satisfied with the revised version.

---

### Review · Reviewer_D6YK · 2022-11-03

**Summary Of Contributions:**

This paper considers the stochastic Douglas-Rachford splitting (SDRS) for general empirical risk minimization (ERM) problems with regularization. It is proved that when a single sample is used per iteration, the convergence rates of SDRS are $O(1/\sqrt{t})$ and $O(1/t)$ under convex and strongly convex settings, with appropriate choices of diminishing step sizes. The authors further study the convergence of mini-batch SDRS where a mini-batch of $p$ samples are used per SDRS iteration and show that the convergence rate stays the same.

**Broader Impact Concerns:**

This is a theoretical paper, so I do not see any concerns on the ethical implications.

**Requested Changes:**

Please address my concerns listed in the weakness section.

**Strengths And Weaknesses:**

Strengths:
The authors provide a solid analysis for the SDRS method in the single same setting. The proof is clear and correct.

Weakness:
The presentation of the mini-batch SDRS method is not clear:
1. How are $\tilde w_t^{(k)}$'s initialized?
2. In line 2 of Algorithm 2, a mini-batch is used, how is this different from the mini-batch freshly sampled in line 3? The conditional expectation used in the analysis requires that $w_{t+1}$ to be independent of the randomness in line 3. So I guess the mini-batch in lines 2 and 3 are different. If so, the notation needs to be changed as it is currently misleading.
2. Do we need to maintain $\tilde w_t^{(k)}$ for $k \in { 1, \ldots, n }$? If not, what is $\tilde w_t^{(k)}$ in line 5?
3. The proof of Theorem 2.7 is incomplete as the authors claim that the omitted part resembles the proof of Theorem 2. I tried to check this but it does not go through trivially. Please provide a complete proof.
4. In Theorem 2.7, the convergence of mini-batch SDRS is exactly the same as the one of SDRS. In that case, what is the point of using a mini-batch of samples per iteration? I would expect certain speed-up that depends on the batch size $p$.

---

> ### Author Response · Authors · 2022-11-07
> **Thank you for your positive assessment**
>
> Regarding your questions about the mini-batch SDRS:
>
> 1. They are initialized randomly at $\tilde{w}_0$. We have revised the description of Algorithm 2 to clarify this.
>
> 2. We were not clear in the first version that the mini-batch SDRS will generate $p+1$ sequences ${\tilde{w}_t^{(1)}}, \ldots, {\tilde{w}_t^{(p)}}$, and ${w_t}$, line 2 computes $w_t$ while line 5 computes each of $\tilde{w}_t^{(k)}$. In line 3, a stochastic function is sampled for each $k$ and a proximal update is computed accordingly; then going to the next iteration, the $p$ stochastic updates are combined together as in line 2. We have clarified it in the revision and also revised the pseudo-code of Algorithm 2.
>
> 3. Yes, we do. Notice that $k=1,\ldots,p$, where $p$ is the batch size, not $n$. As we clarified, the mini-batch SDRS will generate $p+1$ sequences $\{\tilde{w}_t^{(1)}\}, \ldots, \{\tilde{w}_t^{(p)}\}$, and $\{w_t\}$.
>
> 4. We have included the complete proof of Theorem 2.7 in the revision.
>
> 5. Notice that the claim is that the *expected convergence* is the same with mini-batch, and in practice one would see that the variance of the convergence to improve with mini-batch. To illustrate the difference, let us consider the convergence of the subgradient method vs. stochastic subgradient. Following the analysis of John Duchi's monograph *Introductory Lectures on Stochastic Optimization*, for generic convex functions the rates are both $O(1/\sqrt{t})$. The difference is that the convergence of subgradient is deterministic, while that of SGD is in expectation. This means even the widely-used SGD mini-batch converges the same *in expectation*, while the variance is almost always observed to be smaller. Unfortunately, to the best of our knowledge, there has not been analysis about the variance of the convergence of any stochastic algorithm.

---

> > ### Comment · Reviewer_D6YK · 2022-11-19
> > **Response to the author's comments**
> >
> > Thank you for the clarification. I understand now how $\tilde w_0$ is initialized and how the algorithm proceeds with mini-batch:
> > * Mini-batch SDRS runs with $p$ individual "threads", where each thread maintains a variable $\tilde w_t^{(p)}$. The average variable from these $p$ threads is then used to produce the next "global" iterate $w_{t+1}$ with a proximal step.
> >
> > This concept of mini-batch is a bit different from the standard one, and the authors did not provide the theoretical justification of the benefit of the mini-batch approach, which is less than ideal.

---

> ### Comment · Reviewer_D6YK · 2022-11-19
> **Final recommendation**
>
> I can verify that the statement of the algorithms and proofs in the paper are now clear and correct. I am OK with accepting the paper in the current form. However, this paper would be stronger if the benefit of the mini-batch approach over the single sample on is theoretically justified.

---

### Decision · Action_Editors · 2022-11-19

**Recommendation:** Accept as is

**Comment:**

All three reviewers are now satisfied with the revision of the paper. I am also satisfied after seeing that my requested changes, and that of the three reviewers, have been implemented in the revision. Including precise pseudo code of the algorithms, a more practical output (the average iterate) and a proper literature review. As such, and given these methods are of interest to some of TMLR's audience, I now recommend the paper be accepted as is.

**Audience:**

Because the algorithm being studied is the stochastic version of Douglas-Rachford splitting, it can be applied to minimizing empirical risk (ERM) with a regularizor. This problem should be of interest to some of TMLR's audience. In particular this SDRA method is a generalization of the stochastic proximal algorithm, which also some applications in solving particular ERM problems.

**Claims And Evidence:**

The claims are that the papers provides an analysis for the Douglas-Rachford splitting algorithm (SDRA), including a new mini-batching version, and development of several examples or ERM where it can be applied.

---

> ### Author Response · Authors · 2022-11-21
> **Thank you**
>
> We would like to sincerely thank the AE and all three reviewers for their time on reading our paper and the invaluable suggestions to improve the quality of the paper. We have now submitted the camera-ready version.